# DeltaPhi: Physical States Residual Learning for Neural Operators in Data-Limited PDE Solving

**Xihang Yue**[1,2], **Yi Yang**[1,2], **Linchao Zhu**[1,2*]
[1]College of Computer Science and Technology, Zhejiang University
[2]The State Key Lab of Brain-Machine Intelligence, Zhejiang University
https://github.com/yuexihang/DeltaPhi

## Abstract

The limited availability of high-quality training data poses a major obstacle in data-driven PDE solving, where expensive data collection and resolution constraints severely impact the ability of neural operator networks to learn and generalize the underlying physical system. To address this challenge, we propose DeltaPhi, a novel learning framework that transforms the PDE solving task from learning direct input-output mappings to learning the residuals between similar physical states, a fundamentally different approach to neural operator learning. This reformulation provides *implicit data augmentation* by exploiting the inherent stability of physical systems where closer initial states lead to closer evolution trajectories. DeltaPhi is architecture-agnostic and can be seamlessly integrated with existing neural operators to enhance their performance. Extensive experiments demonstrate consistent and significant improvements across diverse physical systems including regular and irregular domains, different neural architectures, multiple training data amount, and cross-resolution scenarios, confirming its effectiveness as a general enhancement for neural operators in data-limited PDE solving.

## 1 Introduction

Due to the lack of analytical solutions, solving complex Partial Differential Equations (PDEs) *e.g.* Navier-Stokes traditionally relies on numerical simulations based on ultra-fine grid division, which consumes expensive computational resources and time. This computational bottleneck has motivated the development of machine learning based PDE solving approaches [1, 2, 3, 4, 5]. In particular, neural operator learning [1, 2, 6] has emerged as a promising direction by directly learning the mapping between input-output function fields, enabling efficient solution prediction through neural network forward computation.

A critical challenge in neural operators is the severe scarcity of high-quality training data [7, 8], which manifests in multiple aspects: (1) collecting comprehensive training data often require expensive physical experiments or high-resolution numerical simulations; (2) computational and storage constraints may restrict the resolution of available data; (3) the collected data frequently exhibits distribution bias due to physical or experimental limitations. These data limitations fundamentally constrain the generalization capability of current neural operators. To address these challenges, we propose DeltaPhi, a novel framework that helps neural operators learn from scarce data.

The core innovation of DeltaPhi lies in reformulating the PDE solving task from learning direct input-output mappings to learning the residuals between similar physical states. This reformulation enables *implicit data augmentation* by leveraging a key property of physical systems: states with similar initial conditions tend to follow similar solutions. By learning residuals between pairs of

---

*Corresponding author.

39th Conference on Neural Information Processing Systems (NeurIPS 2025).

similar trajectories, DeltaPhi effectively expands the diversity of training samples without requiring additional data collection. This is fundamentally different from existing approaches that attempt to learn the straight mapping for each sample independently.

The proposed framework presents two advantages: (1) Enhanced Training Sample Diversity: The framework enables enhanced diversity in training labels through random auxiliary solution sampling, effectively mitigating distribution bias issues. This training distribution augmentation preserves physical validity while expanding the learned solution space. (2) Architecture Flexibility: DeltaPhi is architecture-agnostic and can seamlessly enhance existing neural operators [2, 9, 10] through a unified residual learning mechanism, making it applicable across different PDE solving scenarios.

We conduct extensive experiments to validate the effectiveness of physical states residual learning across various settings: (1) diverse physical systems including both regular domains (Darcy Flow, Navier-Stokes) and irregular domains (Heat Transfer, Blood Flow); (2) different neural architectures including spectral-based and attention-based operators; (3) varied numbers of training samples to test robustness under data scarcity; and (4) cross-resolution scenarios to evaluate generalization capability. The results consistently demonstrate that DeltaPhi significantly improves the performance of base models, particularly in data-limited settings.

Overall, this work introduces a promising framework for enhancing neural operators in practical PDE solving applications where high-quality training data is scarce or expensive to obtain. The physical states residual learning framework could potentially benefit other high-dimensional regression problems in scientific machine learning, especially when dealing with physically governed systems that exhibit stability properties.

## 2 Background and Related Work

### 2.1 Direct Neural Operator Learning

Neural operator learning aims to learn mappings between infinite-dimensional function spaces for solving parametric PDEs [1]. The operator mapping $\mathcal{G} : \mathcal{A} \longrightarrow \mathcal{U}$ maps between Banach spaces $\mathcal{A}$ and $\mathcal{U}$. For steady-state problems like Darcy Flow, the input function $a(x) \in \mathcal{A}$ typically represents coefficients while $u(x) \in \mathcal{U}$ is the solution. For time-series problems like Navier-Stokes, $a(x)$ commonly represents prior states, and $u(x)$ represents future states [2].

Neural operators are constructed using various network architectures, with two prominent families being kernel-based operators and DeepONet-style operators. KernelIntegral-form architectures, such as Graph Neural Operators (GNO) [1], are often based on iterative kernel integration. Deep Operator Networks (DeepONet) [3] instead use separate branch and trunk networks to approximate the operator. These two general forms can be represented as:

$$
\begin{aligned}
\text{KernelIntegral-form:} \quad & \mathcal{G}_\theta = Q \circ \sigma(W_l + \mathcal{K}_l) \circ \cdots \circ \sigma(W_1 + \mathcal{K}_1) \circ P \\
\text{DeepOnet-form:} \quad & \mathcal{G}_\theta(a)(y) = \sum_{k=1}^{p} B_k(a) T_k(y)
\end{aligned}
\tag{1}
$$

In the KernelIntegral-form, $P$ and $Q$ are projection layers, $\mathcal{K}_i$ are learnable kernel integral operators, $W_i$ are local linear operators, and $\sigma$ is a nonlinear activation. In the DeepOnet-form, $B_k(a)$ represents the branch network outputs encoding the input function $a$, and $T_k(y)$ represents the trunk network outputs encoding the output coordinates $y$. It is noted that the framework proposed in this work is applicable to both forms of architectures.

The components of these architectures are differently instantiated in previous works. Fourier Neural Operator (FNO) [2] utilizes the Fourier Transform based integral operation to efficiently learn the physical dynamics using limited training data. Factorized Fourier Neural Operator (FFNO) [10] factorize the Fourier Transform along each dimension, reducing the number of network weights. Clifford Fourier Neural Operator (CFNO) [11] employs the Clifford Algebra in the network architecture, incorporating geometry prior between multi-physical fields. The various neural operators could be simply integrated with the proposed Physical States Residual Learning.

Other works investigate architecture improvements for different needs, including chaotic systems modeling [12], physical-informed instance-wise finetuning [13], accelerating computation [14, 15, 16, 17, 18], spherical fields processing [19], irregular fields learning [20, 21], non-periodic boundary fields

modeling [22], large-scale pretraining [23, 24], *etc.* In addition, some works explore other network backbones, *e.g.* improved frequency-spatial domain transformation [25, 26, 27, 28, 29, 30, 31, 32], convolutions [33], graph neural network [34, 6, 35], attention mechanism [36, 21, 37, 38], and diffusion models [39, 40, 41, 42].

Despite theoretical approximation capabilities, practical limitations persist in generalization across resolutions [13], handling data scarcity [43], and managing biased training distributions [7]. Our work addresses these challenges through residual learning between physical trajectories.

## 2.2 Residual Learning in Previous Works

Some works [44, 45, 46, 47] learn residual between different data samples. Similar to ResMem [44], we also utilize the training set as auxiliary memory during inference, enjoying its generalization enhancement property. The difference lies that (1) we memorize the auxiliary labels and learn the residuals, while ResMem memorizes the residuals and learns the labels, (2) our framework is end-to-end trained while ResMem is trained in separate stages. Similar external memory augmented strategies are popular in the Language Modeling community [48, 49].

Other works learn residuals between low resolution and high resolution representation for image super-resolution [50] and deep learning based CFD simulation [50]. Similar residual learning is investigated in time series prediction task [51, 45]. The primary distinction between these works to ours is that we do not utilize the corresponding low-resolution solutions or previous time-step solutions, but instead, learn the residuals between different trajectories.

Additionally, [52] studies learning residuals between PDE solutions with similar geometric domains. In contrast, our work focuses on a fundamentally different problem of learning residuals between similar physical states, enabling broader applicability. We establish this based on physical system stability and develop implicit data augmentation through similarity-based sampling during training, effectively addressing distribution bias and overfitting issues in data-limited scenarios.

## 3 Methodology

### 3.1 Preliminary: Stability of Physical Systems

**Stability of PDEs.** For a well-posed PDE system with solution operator $\mathcal{G}$, stability means:
$$\|\mathcal{G}(a_1) - \mathcal{G}(a_2)\|_{\mathcal{U}} \leq C\|a_1 - a_2\|_{\mathcal{A}}, \tag{2}$$
where $a_1, a_2$ are input functions and $C$ is a positive constant. This property, also known as Lipschitz continuity of the solution operator, is one of Hadamard's criteria for well-posedness of PDEs. This stability property is well-established in various PDEs: elliptic PDEs satisfy stability under appropriate boundary conditions; parabolic PDEs exhibit stability through their dissipative mechanisms; and even certain chaotic systems maintain stability for suitable time intervals.

It's noted that this stability condition is not an additional assumption imposed by our method, but rather a fundamental prerequisite for *any* data-driven operator learning approach to succeed [3, 2]. If small changes in the input function could lead to arbitrarily large, discontinuous changes in the solution, learning a generalizable mapping would be extremely challenging for neural networks.

**Foundation of Residual Learning.** Stability of PDEs establishes a relationship where the magnitude range of difference between output functions systematically changes with respect to the difference between input functions. This relationship allows us to explicitly control the diversity of output function residuals by selecting input function pairs with varying similarity ranges. This establishes the foundation of residual neural operator learning and enables the effectiveness of *implicit data augmentation*, introduced in Section 3.5.

### 3.2 Residual Operator Mapping: A Unified Formulation

**Residual Operator Mapping**. Based on the stability of physical systems, we formulate the *residual operator mapping* that captures the differences between pairs of physical trajectories:
$$\begin{aligned} \mathcal{G}^{\Delta} &: \mathcal{A}^2 \longrightarrow \Delta\mathcal{U}, \\ \mathcal{A}^2 &: \{(a_i, a_j) \mid a_i \in \mathcal{A}, a_j \in \mathcal{A}\}, \\ \Delta\mathcal{U} &: \{u_i - u_j \mid u_i \in \mathcal{U}, u_j \in \mathcal{U}\}, \end{aligned} \tag{3}$$

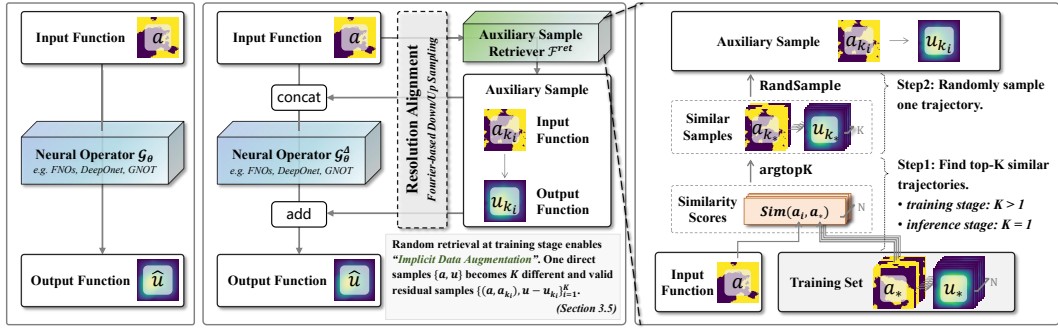

**(a) Direct Neural Operator**   **(b) Residual Neural Operator $\mathcal{G}_\theta^\Delta$**   **(c) Auxiliary Sample Retriever $\mathcal{F}^{ret}$**

Figure 1: The overall architecture of Physical States Residual Learning. Given an *input function* $a_i$, we first sample a similar *auxiliary sample* $(a_{k_i}, u_{k_i})$ from the *training sample set* $\mathcal{T}$. Subsequently, $a_i$ and $(a_{k_i}, u_{k_i})$ are concatenated and fed into the the neural operator $\mathcal{G}_\theta^\Delta$, producing the *predicted states residual*. Finally, the predicted solution $\hat{u}_i$ is obtained by adding the *predicted states residual* with the auxiliary solution $u_{k_i}$.

where $\mathcal{G}^\Delta$ represents the residual mapping operator. $\mathcal{A}^2$ is the Cartesian product of $\mathcal{A}$ with itself, meaning it consists of all possible pairs of functions from $\mathcal{A}$. $\Delta\mathcal{U}$ represents the space of function residuals, which are the differences between pairs of functions from $\mathcal{U}$. $a_*$ and $u_*$ are functions in the space of $\mathcal{A}$ and $\mathcal{U}$, respectively.

**Solving PDEs by Residual Operator $\mathcal{G}^\Delta$.** In PDE solving, given an input function $a_i$, we first retrieve an auxiliary sample $(a_{k_i}, u_{k_i})$ from the set of physical samples with known solution *e.g.* training set. Then the residual neural operator $\mathcal{G}^\Delta$ predicts the solution residual between $u_i$ and $u_{k_i}$. The final solution $u_i$ could be obtained by adding this predicted residual to $u_{k_i}$:

$$u_i = \mathcal{G}^\Delta(a_i, a_{k_i}) + u_{k_i}. \tag{4}$$

We introduce how to obtain auxiliary samples in Section 3.3, and the specific implementation of residual neural operators in Section 3.4.

### 3.3  Auxiliary Sample Retriever $\mathcal{F}^{\mathbf{ret}}$

Given an input function $a_i$ and the training set $\mathcal{T} = \{a_i, u_i\}_{i=1}^N$, we could retrieve an auxiliary sample $(a_{k_i}, u_{k_i})$ from $\mathcal{T}$, formulated as follows:

$$k_i = \mathcal{F}^{\text{ret}}(a_i, \mathcal{T}), \; k_i \in \{1, 2, ..., N\}, \; k_i \neq i, \tag{5}$$

where $\mathcal{F}^{\text{ret}}$ is the sampling function. Below we present our implementation of $\mathcal{F}^{\text{ret}}$ in detail.

**Training stage.** During training, given an input function $a_i$, we randomly retrieve a sample $a_j$ from training set with top-k similarity scores to $a_i$:

$$
\begin{aligned}
\mathcal{F}^{\text{ret}}(a_i, \mathcal{T}) &= \mathtt{RandSample}(\mathtt{argtopK}_{j, j \neq i} \mathtt{Sim}(a_i, a_j)), \\
\mathtt{Sim}(a_i, a_j) &= \frac{a_i \cdot a_j}{\|a_i\| \|a_j\|},
\end{aligned} \tag{6}
$$

where $\mathtt{RandSample}(\cdot)$ represents randomly sampling one element from the given set. $K$ is the sampling range, a hyperparameter that could be adjusted for different PDEs. For a fair comparison, except for specific statements, we set $K = 20$ for all experiments in this work.

**Inference stage.** During inference, given the input function $a^{test}$, we select the most similar auxiliary sample from the training set:

$$\mathcal{F}^{\text{ret}}(a^{test}, \mathcal{T}) = \mathtt{argmax}_j \mathtt{Sim}(a^{test}, a_j). \tag{7}$$

While we take the most similar sample for inference, Appendix A.4 shows that the model is robust to different auxiliary sample choices. When randomly selecting from the top 10 similar samples, repeated testing shows minimal variation - a standard deviation of just 5.41e-5 across five runs. This indicates that neural operators effectively learn to handle varied auxiliary samples for the same input.

**Algorithm 1** Residual Operator Learning (DeltaPhi)

---

**Input:** Training set $\mathcal{T} = \{(a_i, u_i)\}_{i=1}^N$, Neural operator $G_\theta^\Delta$, Number of retrieval neighbors $K$
**Output:** Trained residual operator $G_\theta^\Delta$
**while** not converged **do**
    Sample training pair $(a_i, u_i)$ from $\mathcal{T}$
    Calculate similarities $s_{ij} = \text{sim}(a_i, a_j), a_j \in \mathcal{T}, j \neq i$
    Find indices $\mathcal{N}_i$ of top-$K$ similar samples
    Randomly select $k_i$ from $\mathcal{N}_i$
    Predict residual $\Delta u_i = G_\theta^\Delta([a_i; a_{k_i}])$
    $\hat{u}_i = \Delta u_i + u_{k_i}$
    Update $\theta$ by minimizing $\|\hat{u}_i - u_i\|_2^2$
**end while**

---

**Cross-Resolution Sample Retrieval.** For cross-resolution scenarios where auxiliary samples have different resolutions from the input function, we employ Fourier Transform-based alignment. Specifically, during inference with high-resolution input function $a^H$, we downsample $a^H$ to match training resolution via truncating high-frequency components for similarity calculation:

$$\mathcal{F}_{\text{aligned}}^{\text{ret}}(a^H, \mathcal{T}) = \mathcal{F}^{\text{ret}}(\text{DownSampling}(a^H), a_j), \tag{8}$$

where DownSampling is implemented through frequency-domain truncation:

$$\text{DownSampling}(a^H) = \mathcal{F}_{\text{fourier}}^{-1}(\text{Truncate}(\mathcal{F}_{\text{fourier}}(a^H))). \tag{9}$$

Here $\mathcal{F}_{\text{fourier}}$ and $\mathcal{F}_{\text{fourier}}^{-1}$ denote the Fourier transform and its inverse, respectively. The Truncate operation removes high-frequency components in Fourier space to match the target resolution. This Fourier-based downsampling preserves the low-frequency structures of the physical field while ensuring consistent resolution for similarity computation. This resolution alignment enables effective utilization of auxiliary samples across different resolutions.

**Retrieval Cost Analysis.** The computational overhead of our retrieval process is minimal compared to neural network inference. The total complexity consists of similarity score calculation ($O(N_{field} \cdot N_{train})$) and ranking ($O(N_{train} \cdot \log N_{train})$), where $N_{field}$ is the number of field points and $N_{train}$ is the training set size. Our comprehensive experiments on Darcy Flow demonstrate the high efficiency: GPU-based retrieval only takes 0.2-0.3ms across different training set sizes (100-900), and the entire residual learning pipeline, including both retrieval and data preparation, adds merely 0.5ms to the inference time. This negligible overhead makes our method highly practical for real-world applications. Detailed complexity analysis and timing experiments are provided in Appendix C.2.

## 3.4 Architecture of Residual Neural Operators $\mathcal{G}_\theta^\Delta$

**Residual Neural Operator Backbone.** Benefiting from the universal approximation capability of neural operator networks [2, 10] for direct operator learning, they could be quickly employed as residual neural operators $\mathcal{G}_\theta^\Delta$ for learning residual operator mapping $\mathcal{G}^\Delta$:

$$\mathcal{G}_\theta^\Delta = Q \circ \sigma(W_l + \mathcal{K}_l) \circ \cdots \circ \sigma(W_1 + \mathcal{K}_1) \circ P, \tag{10}$$

where $P$ and $Q$ are projection layers, $\mathcal{K}_i$ and $W_i$ are learnable operators, and $\sigma$ is a nonlinear activation. Next, we introduce how to modify existing neural operators for learning residual mappings.

**Concatenation of Auxiliary Samples.** As aforementioned, the residual operator $\mathcal{G}_\theta^\Delta$ requires additional auxiliary function $a_{k_i}$ as input. In specific implementation, $a_i$ and $a_{k_i}$ could be straightly concatenated along their channel dimension:

$$a_{input} = a_i \oplus a_{k_i}, \tag{11}$$

where $\oplus$ represents the concatenation operation of two vectors. Then the concatenated vector $a_{input}$ is feed into the neural operator network:

$$v_0 = P(a_{input}), \tag{12}$$

where $P$ is the fully connected encoder in the operator network (Equation (10)), and $v_0$ represents the first latent function in the hidden space.

**Concatenation of Additional Information.** The residual operator mapping can be challenging to learn due to the variability of output residuals for different auxiliary samples. To mitigate such difficulty, we introduce additional information related to $a_{k_i}$ as input:

- Taking the corresponding $u_{k_i}(x)$ of $a_{k_i}(x)$ as the input.
- Utilizing calculated relation metrics, *e.g.* similarity between $a_i$ and $a_{k_i}$, as inputs.

Our experimental results (Appendix A.3) indicate that such customized inputs could improve the performance of residual neural operators.

**Physical Residual Connection.** Based on Equation (4), to obtain the final predicted solution $\hat{u}_i$, we introduce a physical residual connection at the final layer of the base operator network. This connection adds the auxiliary solution $u_{k_i}$ to the predicted residual:

$$\hat{u}_i = Q(v_l) + u_{k_i}, \tag{13}$$

where $Q$ is the last fully connected projector of the operator network, $v_l$ is the last latent function in hidden space produced by $\sigma(W_l + \mathcal{K}_l)$ shown in Equation (10). This enables end-to-end training with existing frameworks while maintaining consistent setups (loss functions, trainable parameters, *etc.* ) with the base operator networks.

**Cross-Resolution Sample Integration.** For cross-resolution scenarios where we need to predict high-resolution outputs $u^H$ using low-resolution auxiliary trajectories $(a_{k_i}, u_{k_i})$, we employ Fourier-based upsampling to align the resolutions before integration:

$$a_{input} = \text{UpSampling}(a_{k_i}) \oplus a^H. \tag{14}$$

Similarly, for the residual connection, we upsample the low-resolution auxiliary solution:

$$\hat{u}^H = Q(v_l) + \text{UpSampling}(u_{k_i}), \tag{15}$$

where $Q$ is the decoder and $v_l$ is the final hidden state.

The upsampling is performed through zero-padding in Fourier space to preserve the low-frequency structures while extending to higher resolution, formulated as follows:

$$\text{UpSampling}(a) = \mathcal{F}_{\text{fourier}}^{-1}(\text{ZeroPad}(\mathcal{F}_{\text{fourier}}(a))), \tag{16}$$

where $\mathcal{F}_{\text{fourier}}$ and $\mathcal{F}_{\text{fourier}}^{-1}$ denote the Fourier transform and its inverse, and ZeroPad extends the spectral representation with zeros to match the target resolution. This Fourier-based approach ensures smooth upsampling while preserving the physical characteristics of the auxiliary trajectories.

### 3.5 How DeltaPhi Improves Generalization Capability

In data-limited scenarios, DeltaPhi enhances generalization by mitigating overfitting, which is achieved through a form of *implicit data augmentation* [2] enabled by our residual learning framework.

**Mitigating Overfitting via Implicit Data Augmentation.** The core benefit of residual learning is its effectiveness in mitigating overfitting, particularly in data-limited settings. This is achieved through two complementary mechanisms.

First, DeltaPhi functions as an *implicit data augmentation* strategy. For a training set with $N$ samples, direct learning provides only $N$ input-output pairs. In contrast, DeltaPhi transforms each sample $(a_i, u_i)$ into $K$ distinct residual pairs $\{(a_i, a_k), u_i - u_k\}_{k=1}^{K}$ by pairing it with $K$ different auxiliary samples selected via Section 3.3. The effectiveness of this augmentation stems from the *stability property* of PDEs (Section 3.1), which establishes a structured relationship between input differences and solution residuals. This property allows us to meaningfully control the diversity of the augmented training residuals by sampling auxiliary inputs with varying similarity levels. Therefore, with an opportune retrieval range, we can expand the sparse training distribution—thus mitigating distribution bias—while ensuring the residual learning task remains feasible and does not involve highly dissimilar, difficult-to-learn pairs. This process significantly increases the density and diversity of the training distribution, directly addressing the problem of sparse training data.

---

[2] We name this *implicit data augmentation* because there are no more explicit input-output physical function pairs, but directly training on more diverse residual pairs.

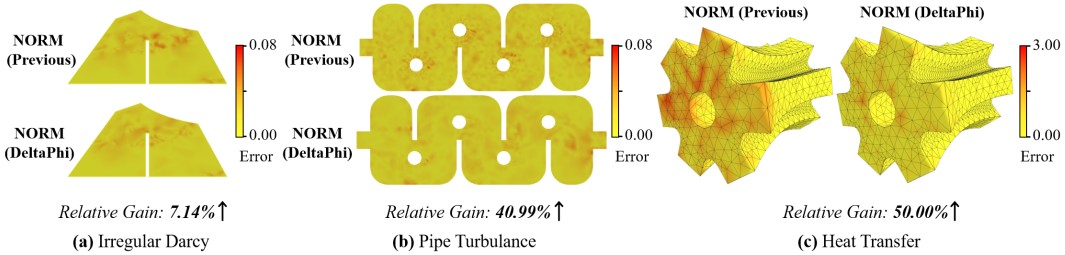

Figure 2: Prediction error visualization on irregular domain problems.

Second, the residual learning formulation itself discourages memorization. Because the predicted residual $\mathcal{G}_\theta^\Delta(a_i, a_{k_i})$ must adapt to different auxiliary samples $a_{k_i}$ for the same input $a_i$, the model is discouraged from simply memorizing a fixed input-output mapping. Thus, the model is less prone to memorizing labels for specific input functions in data-limited scenarios.

From a geometric perspective, this framework effectively shifts the learning paradigm from approximating isolated *points* (absolute solutions) on the solution manifold to learning *tangent vectors* (solution residuals) that describe the manifold's local geometry. Learning these relational vectors provides a richer learning signal. The stability property discussed in Section 3.1 ensures this task is tractable, as it establishes a structured relationship between input differences and solution residuals, allowing us to expand the training distribution while maintaining learnability.

**Tractability of Residual Mapping Learning.** It's noted that learning the residual mapping $\mathcal{G}^\Delta$ can be more difficult than learning the direct mapping $\mathcal{G}$. However, this increased difficulty does not arise from the introduction of spurious high-frequency noise; if two solution functions are smooth, their residual will also be smooth. Instead, the challenge stems from the inherent complexity of the residual mapping itself, which requires the model to synthesize information from two separate input functions ($a_i$ and $a_{k_i}$). Despite this increased complexity, our experiments demonstrate that existing neural operators are fully capable of learning this mapping effectively, even learning to respect its inherent symmetry without explicit constraints (see Appendix A.8).

**Influence of Retrieval Range.** While the retrieval range controls the diversity of residuals for training, our results in Appendix A.1 show DeltaPhi is robust to the selection of this hyperparameter. For experimental consistency, we set $K = 20$ throughout our comparisons. We further analysis its impact across different test splits in Appendix A.2 and discuss its selection in Appendix C.1. In addition, we introduce an alternative sampling strategy without $K$ selection in Appendix A.6.

## 4 Experiment

This section provides the comparison on various PDEs, comparison on resolution generalization problems, and statistical analysis of residual operator learning. We present more experimental results and analysis in Appendix A.

### 4.1 Comparison on Various PDEs

We evaluate residual operator learning across irregular and regular domain PDEs, testing performance with different training data sizes. For fair comparisons, all comparisons use exactly identical hyperparameters, model weights, optimizer settings, and training steps.

**Irregular domain problems.** Following [9], we evaluate five problems: 2D (Irregular Darcy Flow, Pipe Turbulence) and 3D (Heat Transfer, Composite, Blood Flow) scenarios, all defined on irregular domains represented by discrete triangle meshes. We compare against GraphSAGE [53], DeepOnet [3], POD-DeepOnet [54], FNO [2] and NORM [9] baselines, implementing a residual learning version of NORM for direct comparison. Appendix B provides more details.

Results in Table 1 show that NORM (DeltaPhi) outperforms all baselines across irregular domain problems. Performance gains are most notable for Pipe Turbulence and Heat Transfer, showing 40-50% improvement over NORM, demonstrating the residual operator's effectiveness.

Table 1: Relative error comparison of operator learning on irregular domains.

| Model | Irregular Darcy (Train Size=1000) | Pipe Turbulence (Train Size=300) | Heat Transfer (Train Size=100) | Composite (Train Size=400) | Blood Flow (Train Size=400) |
|---|---|---|---|---|---|
| GraphSAGE | 6.73e-2 | 2.36e-1 | - | 2.09e-1 | - |
| DeepOnet | 1.36e-2 | 9.36e-2 | 7.20e-4 | 1.88e-2 | 8.93e-1 |
| POD-DeepOnet | 1.30e-2 | 2.59e-2 | 5.70e-4 | 1.44e-2 | 3.74e-1 |
| FNO | 3.83e-2 | 3.80e-2 | - | - | - |
| NORM (Direct) | 1.05e-2 | 1.01e-2 | 2.70e-4 | 9.99e-3 | 4.82e-2 |
| NORM (DeltaPhi) | **9.75e-3** | **5.96e-3** | **1.35e-4** | **9.18e-3** | **4.29e-2** |
| Relative Gain | 7.14% ↑ | 40.99% ↑ | 50.00% ↑ | 8.11% ↑ | 11.00% ↑ |

In addition, we visualize the prediction error in Figure 2. The error value is calculated as the absolute difference between the predicted function $\hat{u}$ and ground-truth function $u^{GT}$, *i.e.* $|\hat{u} - u^{GT}|$. Compared to previous direct learning methods, DeltaPhi significantly reduces the error scale.

**Regular domain problems.** We conduct the experiment on two regular domain problems *i.e.* Darcy Flow and Navier-Stokes, following [2]. The training data amount is 100. We validate DeltaPhi on different neural operators, including FNO [2], FFNO [10], CFNO [11], GNOT [21], Galerkin [36], MiOnet [55] and Resnet [56]. The metric is relative L2 error. Appendix B presents more details.

Table 2 shows DeltaPhi improves performance across all base models for both Darcy Flow and Navier-Stokes equations, confirming its effectiveness on regular domains. The improvement on the Navier-Stokes, while modest compared to Darcy Flow, is particularly noteworthy given the inherently chaotic nature of fluid dynamics at viscosity $\upsilon = 1e - 5$. Even within this challenging scenario characterized by weakened correlation between similar initial conditions $a(x)$ and their evolved states $u(x)$, DeltaPhi still achieves consistent gains. This robustness to complex dynamics strengthens conclusions regarding the method's general applicability. Performance on temporal systems could be further enhanced through designs such as performing auxiliary sample retrieval at each step of time-series prediction to mitigate the decay in correlation between input and output functions.

Table 2: Relative error comparison on regular Darcy Flow and Navier-Stokes Equation.

| Model | Darcy Flow | | | Navier-Stokes Equation | | |
|---|---|---|---|---|---|---|
| | Direct Learning (Direct) | Residual Learning (DeltaPhi) | Relative Gain | Direct Learning (Direct) | Residual Learning (DeltaPhi) | Relative Gain |
| FNO | 3.70e-2 | 3.31e-2 | 10.54% ↑ | 2.24e-1 | 2.13e-1 | 4.86% ↑ |
| FFNO | 5.22e-2 | 2.93e-2 | 43.76% ↑ | 2.40e-1 | 2.20e-1 | 8.54% ↑ |
| CFNO | 4.79e-2 | 3.15e-2 | 34.23% ↑ | 3.51e-1 | 3.35e-1 | 4.56% ↑ |
| GNOT | 6.74e-2 | 5.50e-2 | 18.39% ↑ | 4.16e-1 | 4.06e-1 | 2.33% ↑ |
| Galerkin | 6.78e-2 | 6.54e-2 | 3.60% ↑ | 3.60e-1 | 3.54e-1 | 1.86% ↑ |
| MiOnet | 8.61e-2 | 8.22e-2 | 4.53% ↑ | 4.79e-1 | 4.61e-1 | 3.80% ↑ |
| Resnet | 1.25e-1 | 1.07e-1 | 14.14% ↑ | 4.39e-1 | 4.30e-1 | 1.87% ↑ |

**Different training scales.** We train residual operators using FNO, FFNO, and Resnet on Darcy Flow with varying training set sizes (100, 300, 500, 700, and 900) and test their performance on the additional 100 samples.

The experimental results are shown in Table 3. On different training set sizes, DeltaPhi improves the prediction perfor-

Table 3: Comparison on different training scales.

| Model | Different Train Size | | | | |
|---|---|---|---|---|---|
| | 100 | 300 | 500 | 700 | 900 |
| FNO (Direct) | 3.70e-2 | 1.40e-2 | 1.02e-2 | 8.37e-3 | 7.39e-3 |
| FNO (DeltaPhi) | 3.31e-2 | 1.34e-2 | 9.64e-3 | 8.06e-3 | 7.18e-3 |
| Relative Gain | 10.54% ↑ | 4.08% ↑ | 5.49% ↑ | 3.74% ↑ | 2.83% ↑ |
| FFNO (Direct) | 5.22e-2 | 1.69e-2 | 9.73e-3 | 7.48e-3 | 6.16e-3 |
| FFNO (DeltaPhi) | 2.93e-2 | 1.11e-2 | 7.20e-3 | 6.22e-3 | 5.30e-3 |
| Relative Gain | 43.76% ↑ | 34.47% ↑ | 25.98% ↑ | 16.79% ↑ | 14.02% ↑ |

mances of neural operators. As the number of available training data decreases, the relative improvement percent (the row "Relative Gain") consistently increases on FNO and FFNO. In addition, several residual neural operators even outperform direct neural operators using less training data amount, *e.g.* FFNO (DeltaPhi) with training size 500 outperforms FFNO (Direct) with training size 700, Resnet (DeltaPhi) with training size 700 outperforms Resnet (Direct) with training size 900. The results empirically demonstrate the effectiveness of DeltaPhi across different training set scales.

## 4.2 Comparison on Resolution Generalization Problem

This section presents the experimental results of training-free resolution generalization.

**Resolution generalization problem.** Fourier neural operator [2] enables zero-shot resolution generalization inference. However, when the training data grid is excessively coarse, the inference

performance over high resolution data drops a lot. The performance degeneration is more serious when learning operator mapping between fields with weaker spatial continuity such as Darcy Flow.

**Setup.** We conduct the training-free resolution generalization experiment on Darcy Flow [2]. Specifically, we train the neural operators with low resolutions $85 \times 85$, $43 \times 43$, $31 \times 31$, and $22 \times 22$ respectively, then evaluate them using high resolution $421 \times 421$ test data. Both FNO [2] and FFNO [10] are compared as the base models. The training set scale includes 100 and 900 trajectories.

Table 4: Relative error comparison on zero-shot resolution generalization.

| Model | Train Size=100 | | | | Train Size=900 | | | |
|---|---|---|---|---|---|---|---|---|
| | $85 \times 85$ | $43 \times 43$ | $31 \times 31$ | $22 \times 22$ | $85 \times 85$ | $43 \times 43$ | $31 \times 31$ | $22 \times 22$ |
| FNO (Direct) | 7.29e-2 | 1.19e-1 | 1.49e-1 | 1.70e-1 | 6.73e-2 | 1.11e-1 | 1.34e-1 | 1.76e-1 |
| FNO (DeltaPhi) | 6.91e-2 | 1.13e-1 | 1.33e-1 | 1.53e-1 | 4.91e-2 | 9.37e-2 | 1.15e-1 | 1.57e-1 |
| **Relative Gain** | 5.13% ↑ | 5.06% ↑ | 10.43% ↑ | 10.08% ↑ | 27.04% ↑ | 15.65% ↑ | 14.29% ↑ | 10.31% ↑ |
| FFNO (Direct) | 6.64e-2 | 1.04e-1 | 1.24e-1 | 1.43e-1 | 4.89e-2 | 1.01e-1 | 1.27e-1 | 1.49e-1 |
| FFNO (DeltaPhi) | 6.34e-2 | 9.90e-2 | 1.18e-1 | 1.36e-1 | 4.42e-2 | 9.00e-2 | 1.12e-1 | 1.35e-1 |
| **Relative Gain** | 4.53% ↑ | 5.06% ↑ | 4.84% ↑ | 5.09% ↑ | 9.48% ↑ | 10.72% ↑ | 11.94% ↑ | 9.29% ↑ |

**Performance comparison.** Table 4 presents the resolution generalization results. Compared to FNO [2], FFNO [10] shows preferable generalization ability for its less number of network weights. In all experimental settings, the proposed physical residual models (tagged with "DeltaPhi") evidently outperform their counterpart base models. The performance improvement of residual operator learning is more noticeable when the training set size is 900. In this setting, FNO (DeltaPhi) managed to match, or even surpass FFNO [10] on resolution $85 \times 85$, $43 \times 43$, and $22 \times 22$, despite using the relatively weaker base network. The results

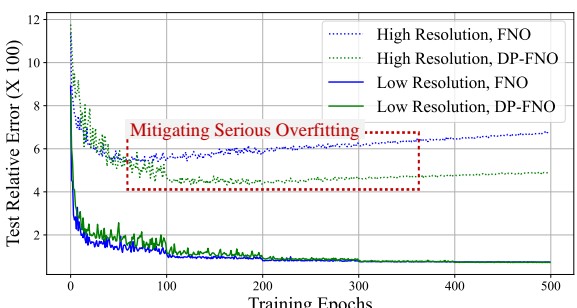

Figure 3: Training curve comparison. "DP-*" denotes the proposed residual learning version of base models.

empirically draw the conclusion that physical residual learning improves zero-shot resolution generalization performance.

**Training curve comparison.** We also report the test loss curve during training in Figure 3. The training amount is 900 and the training resolution is $85 \times 85$. As the training proceeds, the relative error on resolution $85 \times 85$ data continues to decline. However, after some epochs, the loss on resolution $421 \times 421$ gradually increases due to overfitting. The proposed residual operator (named with DP-FNO) significantly mitigates this overfitting tendency of the base model (named with FNO).

### 4.3 Statistical Analysis of Residual Operator Learning

This section validates DeltaPhi's foundations through statistical analysis.

**Similarity between output functions.** We validate the stability property in Section 3.1, which states that similar inputs lead to similar solutions. This property is essential for our residual learning approach. In Figure 4, we show how the normalized distance between output functions $u^{test}(x)$ and $u_{k_t}(x)$ changes as the similarity rank of their input functions increases for Darcy Flow. (Appendix D.1 provides visualization details.) The normalized distance grows consistently

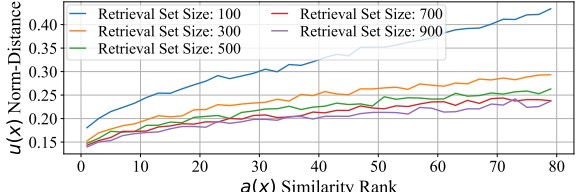

Figure 4: The curve of distance between $u^{test}(x)$ and $u_{k_t}(x)$ as retrieval similarity rank increases.

with similarity rank across all retrieval set sizes. This confirms the stability property - when input functions become less similar, their solutions also become less similar. The relationship also shows that increasing the retrieval range $K$ during training adds more diversity to the training samples by

including a wider range of residual patterns, thereby expanding the learned solution space while maintaining physical validity.

**Training label distribution comparison.** We demonstrate how DeltaPhi provides implicit data augmentation on Darcy Flow by visualizing label distributions. We project both training and testing labels (output functions for direct learning, output residuals for DeltaPhi) into 2D using PCA. Appendix D.2 details this process. Figure 5 reveals the main benefits of our residual learning framework: In direct operator learning (left), limited training data restricts coverage of the solution space, making generalization difficult. This shows the fundamental challenge of distribution bias in data-limited scenarios. With DeltaPhi (right), we see that: (a) the testing label distribution becomes more concentrated, showing that learning residuals is easier than learning absolute solutions, and (b) the training labels show greater spread, demonstrating the implicit data augmenta-

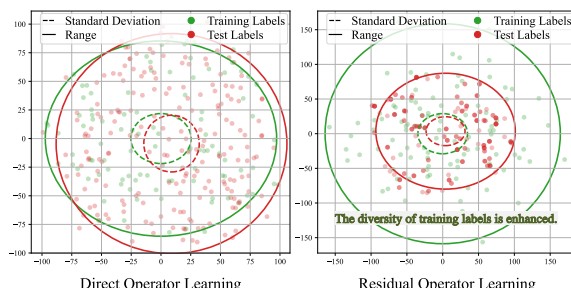

Direct Operator Learning          Residual Operator Learning

Figure 5: Label distribution visualization. The points represent dimension-reduced labels ($u_*(x)$ for direct learning, $u_*(x) - u_{k_*}(x)$ for residual learning) through Principal Component Analysis. The ellipses with dotted lines and solid lines represent standard deviation and range, respectively. Green and red color correspond to training and testing set, respectively.

tion claimed in Section 3.5. By creating diverse valid residual pairs through controlled auxiliary sample retrieval, DeltaPhi effectively enhances the training distribution while preserving physical validity.

## 5 Limitation

Despite the comprehensive validation of DeltaPhi's superiority through an extensive range of experiments, we recognize some inherent limitations that, nevertheless, do not impact the solidity of our conclusions. First, although highly challenging, it is significant to study more realistic applications where training data acquisition is extremely costly, such as high-fidelity fluid modeling or biomedical simulations. Second, we acknowledge the potential constraints of using a general cosine similarity metric. While effective, this metric may not be optimal for all physical systems. This is particularly relevant for highly chaotic systems, such as the low-viscosity Navier-Stokes equations (Table 2), where DeltaPhi, while effective, showed a more modest relative improvement compared to other problems. This suggests that the complexities associated with such systems, where simple initial state similarity may not sufficiently capture the relationship between solution trajectories, warrant further investigation. Future work could explore these aspects, for instance, by developing more advanced auxiliary sample retrieval strategies better suited for complex dynamics, which may unlock greater performance gains in these challenging applications.

## 6 Conclusion

This work proposes to learn physical state residuals for PDE solving by reformulating the task from direct mapping to learning residuals between similar physical states, supported by physical system stability. This approach enables implicit data augmentation without requiring additional data collection, effectively addressing challenges in data-limited scenarios. We validate the effectiveness across various physical systems, neural architectures, and both regular and irregular domains. We hope this framework provides insights for future development of machine learning based PDE solving.

## Acknowledgments

This work is partially supported by National Science and Technology Major Project (2022ZD0117802). This work was supported in part by General Program of National Natural Science Foundation of China (62372403) and "Pioneer" and "Leading Goose" R&D Program of Zhejiang (No. 2025C02032). This work is also supported by the Fundamental Research Funds for the Central Universities (226-2025-00080) and the Earth System Big Data Platform of the School of Earth Sciences, Zhejiang University.

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

# A Additional Experiment

## A.1 Impact of Retrieval Range and Random Sampling

This section analyzes how different retrieval range values ($K$) affect the performance of our residual learning framework. Additionally, we investigate the importance of similarity-based retrieval by replacing it with random sampling (denoted as "w/o $\mathcal{F}^{\text{ret}}$").

Table 5: Relative error comparison of FNO (DeltaPhi) with different retrieval ranges ($K$) and training set sizes on Darcy Flow. The symbols ↑ and ↓ denote performance increasing and decreasing respectively, compared to direct learning.

| Training Set Size | FNO (Direct) | FNO (DeltaPhi) with different $K$ values | | | | | | w/o $\mathcal{F}^{\text{ret}}$ |
|---|---|---|---|---|---|---|---|---|
| | | $K$=5 | $K$=10 | $K$=20 | $K$=30 | $K$=40 | $K$=50 | |
| 100 | 3.57e-2 | 3.34e-2 ↑ | 3.31e-2 ↑ | 3.31e-2 ↑ | 3.31e-2 ↑ | 3.36e-2 ↑ | 3.30e-2 ↑ | 3.04e-2 ↑ |
| 300 | 1.41e-2 | 1.38e-2 ↑ | 1.41e-2 ↑ | 1.35e-2 ↑ | 1.36e-2 ↑ | 1.40e-2 ↑ | 1.37e-2 ↑ | 1.62e-2 ↓ |
| 500 | 1.02e-2 | 1.01e-2 ↑ | 1.01e-2 ↑ | 9.64e-3 ↑ | 9.96e-3 ↑ | 1.00e-2 ↑ | 1.01e-2 ↑ | 1.29e-2 ↓ |
| 700 | 8.37e-3 | 8.29e-3 ↑ | 8.55e-3 ↓ | 8.06e-3 ↑ | 8.04e-3 ↑ | 8.03e-3 ↑ | 8.19e-3 ↑ | 1.10e-2 ↓ |

**Impact of retrieval range $K$.** Table 5 presents the relative error of FNO (DeltaPhi) with different retrieval range values across various training set sizes. We observe several important patterns:

- In data-limited scenarios (100 to 500 samples), DeltaPhi demonstrates robust performance across different $K$ values, obtaining consistent performance gains across diverse selection of $K$.
- With very limited training data (100 samples), larger $K$ values (40-50) tend to perform slightly better, suggesting that expanding the retrieval range becomes beneficial when the training data is extremely sparse.
- Conversely, with abundant training data (700 samples), moderate $K$ values (20-40) generally outperform both smaller and larger retrieval ranges, indicating an optimal balance between sample diversity and learning difficulty.

The consistent performance across different $K$ values demonstrates that our framework is not overly sensitive to this hyperparameter, making it practical for real-world applications where exhaustive hyperparameter tuning may not be feasible.

**Importance of similarity-based retrieval.** The last column of Table 5 shows the performance when auxiliary samples are randomly selected from the training set without considering similarity ("Random Sampling"). Comparing this with similarity-based retrieval reveals:

- For most training set sizes (700, 500, and 300), random sampling performs significantly worse than similarity-based retrieval, with error increases compared to the best $K$ value configuration.
- This performance degradation confirms our discussion in Section 3.1: the stability property of PDEs makes learning residuals between similar states more tractable than between arbitrary states.

These results empirically validate the importance of similarity-based auxiliary sample retrieval in our framework, particularly as the training set size increases. The optimal retrieval strategy may depend on the specific characteristics of the PDE system and the available training data, but similarity-based retrieval provides a robust default approach for most scenarios.

## A.2 Performance Analysis on Different Test Distribution

To evaluate the robustness of DeltaPhi under varying test distributions, we conduct experiments on test samples with different distances from the training set. Specifically, we first compute the cosine similarity score between each test sample and its closest training sample on Darcy Flow. Based on these scores, we divide the test set into 7 non-overlapping splits, where a lower average score indicates a more challenging split due to greater deviation from the training distribution.

We evaluate both direct operator learning and residual operator learning using FNO as the base model across these splits. Additionally, we test different retrieval range settings ($K$) to analyze their impact on performance. The results are presented in Table 7.

The experimental results reveal several key findings:

Table 6: Statistics of different test splits based on similarity to training data.

| | Sample Number | Min Score | Max Score | Average Score |
|---|---|---|---|---|
| Split1 | 23 | 0.85 | 0.86 | 0.86 |
| Split2 | 14 | 0.86 | 0.88 | 0.87 |
| Split3 | 30 | 0.88 | 0.90 | 0.89 |
| Split4 | 37 | 0.90 | 0.91 | 0.90 |
| Split5 | 33 | 0.91 | 0.93 | 0.92 |
| Split6 | 43 | 0.93 | 0.95 | 0.94 |
| Split7 | 16 | 0.95 | 0.96 | 0.95 |

Table 7: Relative error comparison on different test splits. Numbers in parentheses show relative improvement over direct learning.

| Methods | Split1 | Split2 | Split3 | Split4 | Split5 | Split6 | Split7 |
|---|---|---|---|---|---|---|---|
| Direct Learning | 5.08e-2 | 5.00e-2 | 4.71e-2 | 3.31e-2 | 2.77e-2 | 2.73e-2 | 2.43e-2 |
| DeltaPhi ($K$=20) | 4.77e-2 | 4.68e-2 | 4.32e-2 | 3.14e-2 | 2.58e-2 | 2.46e-2 | 2.18e-2 |
| | (6.06%↑) | (6.34%↑) | (8.28%↑) | (5.30%↑) | (6.83%↑) | (10.09%↑) | (10.46%↑) |
| DeltaPhi ($K$=50) | 4.80e-2 | 4.66e-2 | 4.32e-2 | 3.13e-2 | 2.57e-2 | 2.47e-2 | 2.11e-2 |
| | (5.39%↑) | (6.73%↑) | (8.35%↑) | (5.51%↑) | (7.24%↑) | (9.76%↑) | (13.10%↑) |
| DeltaPhi ($K$=5) | 4.83e-2 | 5.01e-2 | 4.39e-2 | 3.27e-2 | 2.52e-2 | 2.43e-2 | 2.10e-2 |
| | (4.78%↑) | (-0.28%↓) | (6.76%↑) | (1.31%↑) | (8.88%↑) | (11.15%↑) | (13.84%↑) |

- DeltaPhi with appropriate retrieval ranges ($K$=20 or $K$=50) consistently outperforms direct operator learning across all splits, even on the most challenging ones (Split1, Split2, Split3) that significantly deviate from the training distribution.

- The choice of retrieval range $K$ has a notable impact on performance. A relatively small $K$ value ($K$=5) leads to smaller improvements on challenging splits (Split1, Split2, and Split4), and can even underperform direct learning in some cases (Split2).

- $K$ values ($K$=20 or $K$=50) in appropriate ranges generally provide stable improvements across different splits, suggesting they enable more robust generalization under distribution shifts.

These results demonstrate that DeltaPhi maintains its effectiveness even when test samples significantly differ from training data, supporting its robustness in practical scenarios where training and testing distributions may not perfectly align.

### A.3   Customized Auxiliary Input Ablation

This section conducts ablation experiments on the customized auxiliary input.

**Setup.** We evaluate the influence of customized auxiliary input on Navier-Stokes. The training trajectory number is 100. FNO [2] is used as the base model. We test 3 types input of $a_{k_i}(x)$, including complete $a_{k_i}(x)$, empty $a_{k_i}(x)$ and partial (only input last 3 trajectory steps) $a_{k_i}(x)$. In addition, we experiment with the inclusion or omission of $u_{k_i}(x)$, as well as the addition or exclusion of $Score_{i,k_i}$.

**Customized Auxiliary Input Influence.** The results are shown in Table 8. The first three rows indicate that utilizing partial $a_{k_i}(x)$ yields the best results for $a_{k_i}(x)$. When comparing the third row with the fourth row, or the fifth row with the sixth row, it's evident that incorporating $Score_{i,k_i}$ is also meaningful. Additionally, when comparing the third row with the fifth row, as well as the fourth row with the sixth row, the inclusion of $u_{k_i}(x)$ proves to be extremely crucial. The above findings suggest that the utilization of customized input is beneficial, thereby confirming the conclusion that proper customized input could improve performance.

### A.4   Auxiliary Sample Selection during Inference

As described in Section 3.3, we greedily select take the auxiliary sample with the top similarity score during inference. To validate the robustness of the trained residual neural operator on different

Table 8: Ablation of customized auxiliary input on Navier-Stokes. **Bold font**, underline and ~~wavyline~~ respectively denote the best, second best, and inferior results.

| Customized Auxiliary Input | | | Relative Error |
|---|---|---|---|
| $a_{k_i}(x)$ | $u_{k_i}(x)$ | $Score_{i,k_i}$ | |
| All | ✓ | ✓ | 2.14e-1 |
| ✗ | ✓ | ✓ | 2.20e-1 |
| Partial | ✓ | ✓ | **2.13e-1** |
| Partial | ✓ | | 2.13e-1 |
| Partial | | ✓ | ~~2.34e-1~~ |
| Partial | | | ~~2.37e-1~~ |

auxiliary samples, we experiment with randomly sampled auxiliary trajectories from the top-10 similarity scores. The experiment is conducted on Darcy Flow with FNO as the base model.

Table 9: Results of repeat Evaluation using random auxiliary sample with Top-10 similarity score.

| | Relative Error of FNO (DeltaPhi) |
|---|---|
| Infer 1 | 3.3441876e-2 |
| Infer 2 | 3.3364178e-2 |
| Infer 3 | 3.3501464e-2 |
| Infer 4 | 3.3489122e-2 |
| Infer 5 | 3.3489122e-2 |
| **STD** | 5.40649e-5 |

Table 9 presents the relative error of every experimental run and the standard deviation (STD) across five repeated experiments. The low standard deviation value **5.41e-5** indicates that the trained residual neural operators are not sensitive to the selection of auxiliary samples. This concludes that the neural operator effectively learns the proposed residual operator mapping.

## A.5   Different Similarity Function

We use Cosine Similarity for its two advantages:

- Unaffected by the magnitude of the numerical value. Thus it could be conveniently applied to various PDEs without considering the numerical scale distribution of physical fields.
- Taking normalized values in $[0, 1]$. We could simply use it as an additional network input. As Table 8 shows, incorporating the similarity scores improves the model's performance on the Navier-Stokes equation.

To explore more retrieval metrics for scientific machine learning, we experiment with other discrepancy metrics (all the distance values are calculated after normalization), as shown in Table 10.

Table 10: Results of DeltaPhi-FNO using different similarity functions.

| Similarity Functions | Relative Error |
|---|---|
| Cosine Similarity | 3.31e-2 |
| Euclidean Distance | 3.31e-2 |
| Manhattan Distance | 3.28e-2 |

The results indicate that different distance functions yield subtle impacts. Euclidean Distance and Manhattan Distance are also appropriate choices. Additionally, inspired by the great success of the NLP community, conducting retrieval in the latent space may yield superior results. We will explore this in future work.

## A.6 Alternative Sampling Strategy: Importance Sampling

In addition to the similarity-based sampling strategy with a fixed range $K$ described in Section 3.3, we also explored the importance sampling approach. Instead of selecting auxiliary samples from a fixed-size neighborhood, this method assigns sampling probabilities to all training instances based on their similarity scores with the input function.

Specifically, given an input function $a_i$, the probability of selecting any training sample $a_j$ as the auxiliary sample is defined as:

$$P(a_j|a_i) = \frac{\exp(\text{sim}(a_i, a_j)/\tau)}{\sum_{k=1}^{N} \exp(\text{sim}(a_i, a_k)/\tau)}, \tag{17}$$

where $\text{sim}(\cdot, \cdot)$ is the cosine similarity function defined in Equation (6), $\tau$ is a temperature parameter controlling the sharpness of the distribution, and $N$ is the total number of training samples.

We evaluated this importance sampling strategy on two representative problems: Darcy Flow with FNO and Heat Transfer with NORM. The results are presented in Table 11.

Table 11: Performance comparison of different sampling strategies

| Method | Heat Transfer | Darcy Flow |
|---|---|---|
| Baseline (Direct Learning) | 2.70e-4 | 3.70e-2 |
| DeltaPhi ($K = 20$) | **1.35e-4** | 3.31e-2 |
| DeltaPhi (Importance Sampling) | 2.02e-4 | **3.22e-2** |

The experimental results demonstrate that importance sampling can effectively enhance model performance, achieving 25.04% and 12.95% relative improvement on Heat Transfer and Darcy Flow respectively. Notably, on Darcy Flow, importance sampling even outperforms the fixed-range sampling strategy ($K = 20$). This suggests that adaptive probability-based sampling could be a promising alternative to the fixed-range approach, particularly for problems where the similarity structure of the solution space is more complex.

## A.7 Comparison of Interpolation Methods for Cross-resolution Residual Learning

In our framework, we employ Fourier-based interpolation for cross-resolution alignment and integration (as discussed in Section 3.3 and Section 3.4). This choice is advantageous as zero-padding in the frequency domain, which is equivalent to ideal sinc interpolation in the spatial domain, effectively upsamples the signal while exactly preserving the original low-frequency components without distortion. This characteristic aligns with the core principles of FNO-like architectures for zero-shot resolution generalization, which rely on accurately capturing the low-frequency dynamics of the physical solution.

Table 12: Comparison of different interpolation methods on Darcy Flow.

| Interpolation Method | Relative Error |
|---|---|
| Fourier Interpolation | **6.91e-2** |
| Bilinear Interpolation | 7.36e-2 |
| Nearest Interpolation | 7.39e-2 |

To validate this choice, we conducted a comparison with other common interpolation methods, specifically Bilinear and Nearest Interpolation. The experiment was performed on the Darcy Flow problem, training on a $85 \times 85$ resolution and testing on a $421 \times 421$ resolution. The results, presented in Table 12, show that Fourier-based interpolation achieves the lowest error, confirming its suitability for zero-shot generalization tasks.

## A.8 Symmetry Analysis of Residual Operator

A key property of our residual learning formulation is the inherent symmetry between paired samples. To investigate whether the network naturally captures this property without explicit enforcement, we analyze the symmetry behavior of trained residual neural operators.

**Symmetry Metric.** We introduce a Symmetry Loss metric to quantify how well the learned operator respects the expected symmetry for paired samples $(a_i, u_i)$ and $(a_{k_i}, u_{k_i})$:

$$
\text{Symmetry Loss} = \frac{\|\mathcal{G}_\theta^\Delta(a_i, a_{k_i}) + \mathcal{G}_\theta^\Delta(a_{k_i}, a_i)\|_2}{\|\mathcal{G}_\theta^\Delta(a_i, a_{k_i})\|_2} \tag{18}
$$

This metric measures the normalized discrepancy between residual predictions when the input order is reversed. A perfectly symmetric operator would yield a value of zero, while larger values indicate asymmetric behavior.

**Experimental Setup.** We track this metric throughout the training process for our DeltaPhi-FNO model on the Darcy Flow problem. Importantly, we do not explicitly optimize for symmetry in the training objective; we simply observe how the standard training process affects symmetry properties.

Table 13: Evolution of Symmetry Loss during training of DeltaPhi-FNO on Darcy Flow

| Metric | Epoch 0 | Epoch 300 | Epoch 600 | Epoch 900 | Epoch 1200 | Epoch 1500 | Epoch 1800 |
|---|---|---|---|---|---|---|---|
| Symmetry Loss | 2.01e+0 | 1.16e-1 | 8.00e-2 | 4.52e-2 | 4.12e-2 | 3.84e-2 | 3.22e-2 |

**Results and Analysis.** The results in Table 13 demonstrate that the model progressively learns to respect the inherent symmetry of the residual mapping without explicit guidance. Starting from a highly asymmetric state (Symmetry Loss of 2.01), the model naturally converges toward increasingly symmetric behavior, reaching a much lower Symmetry Loss of 3.22e-2 by the end of training.

This emergence of symmetry-aware behavior validates that our residual learning approach inherently captures physically meaningful properties of the underlying systems. The network effectively learns that the relationship between two physical states should maintain consistent properties regardless of which state is considered the reference, aligning with fundamental principles of physical systems.

These findings suggest potential future improvements through explicit symmetry enforcement during training or through specialized architectures designed to leverage this property, which could further enhance the physical consistency and sample efficiency of neural operator learning.

## A.9 Integration with Other Base Models

The formulated residual mapping is a general operator learning task, fundamentally independent of the specific base architecture. Consequently, any neural operator can be employed to learn this mapping. In Table 14, we include two more base models, Transolver [38] and HPM [25], on the Irregular Darcy problem.

Table 14: Performance of DeltaPhi integrated with more base architectures.

| Model | Relative Error |
|---|---|
| Transolver (Direct) [38] | 8.54e-3 |
| Transolver (DeltaPhi) | **8.18e-3** |
| HPM (Direct) [25] | 7.39e-3 |
| HPM (DeltaPhi) | **6.67e-3** |

The results show that DeltaPhi consistently improves the base models. In the future, it is significant to integrate DeltaPhi with more advanced architectures.

# B Experimental Detail

## B.1 Dataset

### B.1.1 Regular Domain

**Darcy Flow.** Darcy Flow is a steady-state solving problem. We conduct the experiment on Darcy Flow using the same dataset as [2], consisting of $421 \times 421$ resolution fields with Dirichlet boundary. The low resolution data *e.g.* $22 \times 22$ are obtained via uniformly downsampling operation.

**Navier-Stokes.** Navier-Stokes is a challenging time-series solving problem. We use the public dataset from [2]. The viscosity of trajectories is $1e-5$ ($Re = 2000$). The spatial resolution is $64 \times 64$. The input and output time step length are both $10$.

### B.1.2 Irregular Domain

For these irregular domain problems, we take the exact same setting with NORM, more details about the dataset can be found in [9].

**Irregular Darcy.** The irregular Darcy problem is to solve the Darcy Flow equation defined on an irregular domain. The input function $a(x)$ represents the diffusion coefficient field and the output function $u(x)$ is the corresponding pressure field. The domain is represented with a set of triangle meshes consisting of 2290 nodes. Following [9], we train the neural operators with 1000 data and test them on additional 200 trajectories.

**Pipe Turbulence.** Pipe Turbulence is a dynamic fluid system described by the Navier-Stokes equation. The computational domain is an irregular pipe shape represented as 2673 triangle mesh nodes. In this problem, the neural operator is required to predict the velocity field in the next frame given the previous velocity field. Same as [9], 300 trajectories are employed for training, and 100 data are used for evaluation in our experiments.

**Heat Transfer.** In the Heat Transfer problem, the energy transfer phenomena due to temperature difference is studied. The system evolves under the governing law described by the Heat equation. The neural operator is optimized to predict the 3-dimensional temperature field in 3 seconds given the initial boundary temperature state. The output physical domain is represented by triangle meshes with 7199 nodes. We use 100 data for training and the rest of 100 data for evaluation.

**Composite.** Composite problem is to predict the deformation field in high-temperature stimulation, which is greatly significant for composites manufacturing. The learned operator is expected to predict the deformation field given the input temperature field. Following [9] The studied geometry in this work is an air-intake structural part of a jet, which is composed of 8232 nodes. The training data size is 400 and the test data size is 100.

**Blood Flow.** This problem aims to predict blood flow in the aorta, which contains 1 inlet and 5 outlets. The blood flow is simulated as a homogeneous Newtonian fluid. The computational domain is completely irregular and represented by a set of triangle meshes comprising 1656 nodes. The simulated temporal length is 1.21 seconds with the temporal step length of 0.01 seconds. The neural operator predicts the velocity field at different times given the velocity boundary at the inlet and pressure boundary at the outlet. In this problem, we have 400 data for training and 100 data for testing, same as [9].

## B.2 Base Model

**Fourier Neural Operator (FNO) [2].** Fourier Neural Operator (FNO) [2] utilizes the Fourier Transform based integral operation to implement the neural operator kernel $\mathcal{K}_i$. We implement the FNO with the officially published code under the up-to-date deep learning framework. All model hyperparameters except the hidden channel width ($64$ in our experiment, $32$ in the original implementation) are kept the same as in the original manuscript. The incremental hidden channels enhance the model's representation capability, obtaining consistent performance improvement in all settings. The optimization setup (Adam [57] optimizer with initial learning rate $0.001$ and weight decay $0.0001$, StepLR scheduler with step size 100 and $\gamma = 0.5$) is consistent with official implementation.

**Factorized Fourier Neural Operator (FFNO) [10].** Factorized Fourier Neural Operator (FFNO) [10] conducts the Fourier Transform along every dimension independently, reducing the model size and enabling deep layer optimization. We implement FFNO via the independent Fourier Transform along each dimension, the improved residual connections, and the FeedForward-based encoder-decoder. All model hyperparameters and optimization setups are consistent with FNO.

**Clifford Fourier Neural Operator (CFNO) [11].** Clifford Fourier Neural Operator (CFNO) [11] employs the Clifford Algebra in the neural network architecture, incorporating geometry prior between multiple physical fields. We employ the official implementation for the model architecture. The signature is set as $(-1, -1)$ and we pad lacked physical channels with zero. All hyperparameters except the channel width (32 in CFNO) remain consistent with FNO. The training setup (including the optimizer and scheduler) is the same as FNO.

**Galerkin [36].** Galerkin proposes to learn the neural operator with linear attention. We employ the same model architecture as the original work. The network depth is 4, the head number in the attention layer is set as 2, and the latent dimension in the feedforward layer is 256. We use the Galerkin attention mechanism and set the dropout value as 0.05. In our experiment, we use the same training setup as FNO.

**GNOT [21].** GNOT is also an attention-based neural operator structure, which introduces the heterogeneous normalized attention layer to handling various inputs such as multiple system inputs and irregular domain meshes. We employ the official architecture implementation. The network depth is set as 4, the hidden dimension is 64, and the activation function is GeLU. The training setup is the same as FNO.

**MiOnet [55].** MiOnet [55] is an enhanced version of DeepOnet [3]. It processes multiple inputs by utilizing multiple branch networks and then merging all branch outputs with several fully connected layers. In our implementation, the depth of all branch layers, trunk layer, and merging layer are set as 4. The latent dimension is set as 128. We take the same training setup with FNO.

**Resnet [56].** Resnet [56] is a classical network architecture in image processing. Although it fails in zero-shot resolution generalization inference for learning infinite dimension operator mapping, the powerful capability of capturing local details deserves attention. We implement Resnet by simply replacing the spectral convolution in FNO with $3 \times 3$ spatial convolution operation. All model hyperparameters and optimization setups remain consistent with FNO.

### B.3 Implementation Details

Except for specific statements, the experimental details are as follows: (a) Retrieval: For training, the auxiliary sample retrieval range $K$ is set as 20. During inference, the auxiliary sample with the most similarity with $a_i$ is retrieved. (b) Input: The customized input function is partial (last 3 channels) $a_{k_i}$ and complete $u_{k_i}$. (c) Optimizing: We use the Adam [57] optimizer with initial learning rate $0.001$ and weight decay $0.0001$. The StepLR scheduler with step size 100 and $\gamma = 0.5$ is used to control the learning rate. The batch size is set as 8. All models are trained for 500 epochs. All experiments could be conducted on a single NVIDIA GeForce RTX 4090 device.

### B.4 Metric

The evaluated metric is Relative L2 Error, defined as:

$$L2 = \frac{1}{N} \sum_{i=1}^{N} \frac{\|\hat{u}_i - u_i\|_2}{\|u_i\|_2}, \tag{19}$$

where $\hat{u}_i$ and $u_i$ are predicted solution and real solution respectively.

## C  Additional Method Detail

### C.1  Selection of $K$ Value

The optimal value of $K$ is hard to determine. It depends on the generalization problem type (limited training data problem and resolution generalization problem in this work), the base models, etc.

Empirically, the larger the value of K, the better the model's generalization range, but the learning difficulty also increases. A proper value of $K$ should ensure enough diversity of training residuals for generalizing to the targeted pending solved physical trajectories. Thus, it is advisable to set a larger K value when the discrepancy between the training set and test set is evident, and the model's representation ability is considerably strong. Through extensive experiments, we find $K = 20$ is an applicable value for the focused generalization problem of this work, ie. the limited amount and resolution of training data.

For more challenging generalization problems (eg. serious data bias problems in realistic scenarios), we provide an empirical selection algorithm for the "initial value" decision of $K$ (the "final value" of $K$ is also related to the base model's capability). Given the training set (noted as $\{a_i^{train}\}_{i=1}^{N_{train}}$), suppose the input functions (noted as $\{a_i^{test}\}_{i=1}^{N_{test}}$) of pending solved physical trajectories is available. The initial value of $K$ could be calculated with the following steps: (1) Retrieve the most similar sample (noted as $\{a_{k_i}^{test}\}_{i=1}^{N_{test}}$) from the training set for every pending solving function. We denote the similarity values between $a_i^{test}$ and $a_{k_i}^{test}$ as $s_i^{test}$. (2) Calculate the maximum value $s_{max}^{test}$ of $\{s_i^{test}\}_{i=1}^{N}$. (3) For every training sample $a_i^{train}$, calculate its similarity with other training trajectories. Denote the index of $r$-th similar sample with $a_i^{train}$ as $k_i^r$, and the similar score between $a_i^{train}$ and $a_{k_i^r}^{train}$ as $s_i^r$. (4) The $K$ is calculated as the minimum $r$ satisfying $s_{max}^{test} \leq s_i^r$ for any $i \in [1, N^{train}]$.

## C.2 Complexity Analysis of Auxiliary Sample Retrieval

The computational cost for retrieval is negligible compared to neural network inference. Here we provide both theoretical complexity analysis and comprehensive experimental validation.

**Theoretical Analysis.** The computational complexity primarily depends on two factors: the training set size (denoted as $N_{train}$) and the number of physical field points (i.e., the resolution of physical field) (denoted as $N_{field}$). The retrieval process consists of two main steps:

- Step 1: Similarity Score Calculation. For each sample in the retrieval set, computing the similarity metric (e.g., Cosine Similarity) has linear time complexity $O(N_{field})$. Thus, calculating similarity scores for all retrieved samples has complexity $O(N_{field} \cdot N_{train})$.
- Step 2: Similarity Score Ranking. Using QuickSort to rank samples based on similarity scores has complexity $O(N_{train} \cdot \log N_{train})$.

The overall time complexity is $O(N_{field} \cdot N_{train} + N_{train} \cdot \log N_{train})$. With typical values of $N_{field}$ and $N_{train}$ ranging from $10^3$ to $10^4$ and $10^2$ to $10^4$ respectively, the computation is feasible on standard hardware.

**Experimental Validation.** We conducted extensive experiments to measure the actual computational overhead of retrieval:

- **Retrieval Time Analysis:** We tested retrieval time across different training set sizes on both CPU and GPU (single RTX 3090) for Darcy Flow ($85 \times 85$ resolution):

Table 15: Retrieval time (in seconds) for different training set sizes

| Device | 100 | 300 | 500 | 700 | 900 |
|--------|------|------|------|------|------|
| CPU | 2.67e-3 | 5.57e-3 | 9.05e-3 | 1.32e-2 | 1.66e-2 |
| GPU | 1.89e-4 | 1.83e-4 | 1.64e-4 | 2.43e-4 | 3.06e-4 |

- **End-to-End Inference Time:** We compared the total inference time per example between direct learning and residual learning across different models:

Table 16: Average inference time (in seconds) comparison

| Method | FNO | FFNO | Galerkin |
|--------|------|------|----------|
| Direct Learning | 1.20e-3 | 2.39e-3 | 5.01e-3 |
| Residual Learning | 1.67e-3 | 2.85e-3 | 5.58e-3 |
| *Increased Time* | 4.67e-4 | 4.66e-4 | 5.75e-4 |

The experimental results demonstrate that:

- GPU-based retrieval is highly efficient, taking only 0.2-0.3ms across different training set sizes.
- The total additional time for residual learning, including retrieval and data preparation, is consistently around 0.5ms.

These comprehensive experiments confirm that the auxiliary sample retrieval introduces negligible computational overhead and does not significantly impact the practical deployment of our method.

### C.3 Discussion on Boundary Conditions

In this work, DeltaPhi does not explicitly process boundary conditions as a separate input. Instead, it implicitly handles them by leveraging the full auxiliary solution $u_{k_i}$ as a strong physical prior. This auxiliary solution, which is a key input to the model, already satisfies the boundary conditions of its corresponding system. The model's effectiveness in handling complex boundary effects is demonstrated by its strong performance on irregular domain problems, such as Pipe Turbulence and Blood Flow, which feature complex inlet/outlet conditions. While our current approach is effective, future work could explore enhancing the retrieval mechanism with a boundary-aware similarity metric for enhanced residual neural operator.

## D   Visualization Analysis Detail

### D.1   Similarity between $u(x)$ Visulization Detail

We visualize the correlation between $u(x)$ normalized distance and $a(x)$ similarity rank (shown in Figure 4) as following steps: Firstly, taking the training set $\mathcal{T}$ as retrieval set, we retrieve auxiliary sample $(a_{k_{i,r}}, u_{k_{i,r}})$ with $r$-th top similarity score for every test sample $(a_i^{test}, u_i^{test})$ in the test set. Here $r$ represents the similarity rank (higher $r$ indicates lower similarity), and we take gradually increasing values from $1$ to $80$ for $r$. Next, we calculate the normalized distance between $u_i^{test}$ with its auxiliary sample $u_{k_{i,r}}$. The normalized distance is defined the same as the relative error in Equation (19).

### D.2   Label Distribution Visualization Detail

We visualize the label distribution (as shown in Figure 5) by reducing the high-dimension function fields to 2 dimensions utilizing Principle Component Analysis. Specifically, for both direct operator mapping and residual operator mapping, we first optimize a PCA model on the training labels (100 labels) and then calculate the dimension-reduced labels for every sample in the training and testing set. Finally, we visualize every two-dimension label point on the graph as well as the ellipses representing the stand deviation and the range along each dimension.

