# OpenReview forum: "DeltaPhi: Physical States Residual Learning for Neural Operators in Data-Limited PDE Solving"
_NeurIPS.cc/2025/Conference — NeurIPS 2025 poster_

### Official Review · Reviewer_cdGa · 2025-06-04

**Clarity:** 3
**Significance:** 3
**Originality:** 3
**Rating:** 4
**Confidence:** 4

**Summary:**

This paper presents DeltaPhi, a novel framework for enhancing neural operator learning in data-scarce PDE solving tasks. DeltaPhi redefines the learning objective from direct input-output mapping to residual prediction between similar physical states. This reformulation leverages the stability properties of physical systems to enable implicit data augmentation without additional data collection.

**Questions:**

1. Have you considered using the auxiliary solution function as an input and directly predicting the target solution, instead of learning the residual? This would allow the model to use the auxiliary data as a form of guidance—conceptually similar to in-context or few-shot learning in large language models, where context from similar examples is used to guide predictions.

2. How does DeltaPhi perform on Navier-Stokes with higher viscosity, e.g. 1e-3 from FNO?

3. In table 4, it seems that the performance gain is larger with more training data, which is the opposite of the normal setting, can you give some reasons?

4. What is the training and testing resolution for Darcy flow in Table 2? As you write following FNO (line 242), I assume the training and testing resolutions are all $85\times 85$?

5. Why is DeltaPhi only applied to one model in Table 1? Is it compatible with all models? All other reported errors has no meaning at all except for NORM.

Minor issues:

1. Do you only refer to kernel integral operators as neural operators? Equation (1) only describes kernel integral operators, it does not describe many other operators, e.g. DeepONet you use in your work.

**Ethical Concerns:**

["NO or VERY MINOR ethics concerns only"]

**Final Justification:**

Overall, my concerns have been addressed. After reading the other reviewers' comments and giving the paper careful consideration, I have decided to keep my score as is, primarily for the following two reasons:

1. The use of cosine similarity is limiting (I read the responses; however, I am not fully convinced), and there is no principled way to identify suitable similarity measures.

2. While I agree with the authors that chaotic systems are challenging for all neural operators, they may pose even greater difficulty for the proposed method. Regarding the results provided: 1e-3 is not chaotic at all. It represents an overly simple and smooth dataset, which favors models like FNO and its variants that are designed to operate on low-frequency signals. Also from your results, we can already see that the performance gain from less smooth dataset (1e-4) is more marginal compared to that from smooth dataset (1e-3).

**Limitations:**

yes

**Quality:**

3

**Strengths And Weaknesses:**

# Strengths
1. The idea of residual learning is very interesting.
2. DeltaPhi is a plug-and-play module, compatible with diverse operator networks, making it highly flexible.
3. Overall, the paper is easy to follow, and necessary details are clearly explained.

# Weaknesses

1. Using cosine similarity to measure the similarity between two functions is potentially problematic. For example, let's say you have two functions $u_1(x)=a$ on a small disk $D_a \subset[0,1]^2$, zero elsewhere, and $u_2(x)=b$ on a different small disk $D_b$, disjoint from $D_a$, no matter what $a$ and $b$ is, the cosine similarity will always be 0. It is definitely problematic because suppose they are initial heat sources in a heat equation, if $a=b=0$, there is no residual at all. If $a$ is very different from $b$, the residual is large. Potentially, $L_2$ or cross-correlation might even be a better choice. However, I do believe the authors need to elaborate more the choice of the similarity metric.

2. All the benefits of DeltaPhi is described verbally without any theoretical foundation.

3. Although the implicit data augmentation adds more training pairs, we do not get any additional information from the training dataset. Also, I think the residuals can be, under some cases, more difficult to learn than the solutions. For example, even if the solutions are very smooth, the residuals may contain sharp transitions; high frequency components can dominate. This is harmful for the learning of neural networks, as well as neural operators. This is potentially the reason why the performance gain on the Navier-Stokes equation is minimal.

4. This can be very subjective, but the performance gain is not very significant, especially for the Navier-Stokes equation on Table 2. The results suggest that the method might struggle with chaotic systems (e.g., Navier-Stokes at low viscosity).

---

> ### Author Rebuttal · Authors · 2025-07-31
>
> Thanks for your thoughtful reviews and insightful questions. They have significantly improved this work. We respond point-by-point below.
>
> > **[W1] Concerns on Cosine Similarity.**
>
> Please see our response in `[W5]` to Reviewer `7n6g`.
>
> > **[W2] Questions on theoretical explanation.**
>
> We understand your concern in the presentation of core benefits of DeltaPhi. However, we wish to highlight that proving theoretical guarantees for generalization in deep learning is notoriously challenging. Our explanation in Section 3.5 for DeltaPhi's benefit is intuitive yet powerful: **it effectively mitigates the overfitting in data-driven neural operators**, particularly in data-limited settings. This is because the model is less prone to memorizing labels when learning residuals.
>
> While a rigorous theoretical analysis remains challenging, DeltaPhi can be understood from a geometric perspective on the solution manifold. The residual learning framework effectively **shifts the learning paradigm from approximating isolated "points" (absolute solutions) to learning "tangent vectors" (solution residuals) that describe the solution manifold's local geometry**.  Learning these "relational vectors" is more powerful because it provides a richer learning signal and thus enables a form of implicit data augmentation.
>
> We will enhance the discussion about benefits of DeltaPhi in our final manuscript based on your helpful suggestion.
>
> > **[W3] Questions on the learning difficulty of residual mapping and the benefits of DeltaPhi.**
>
> First, we clarify that our claim is not that DeltaPhi constitutes a "simplification hypothesis" (mentioned by Reviewer 7n6g). We agree that learning residual mapping can be more challenging. However, we highlight that this difficulty arises from the inherent complexity of the residual mapping itself—requiring the model to synthesize information from two input functions—**rather than from the introduction of harmful or meaningless high-frequency components**. If two solution functions are smooth, their residual will naturally be smooth, not introducing unnecessary sharpness (concerned by Reviewer cdGa). In addition, every training pair strictly adheres to the residual mapping, representing the exact difference between the solutions given two input functions.
>
> Despite the increased learning difficulty, our results demonstrate that **existing neural operators are fully capable of learning residual mapping** under original parameter and training configurations. Notably, the model even learns to implicitly respect the symmetry of the residual mapping without any explicit constraint, as detailed in Appendix A.7.
>
> The core benefit of residual learning is its effectiveness in mitigating overfitting in data-limited scenarios. This is achieved in two aspects: (a) Because the predicted residual must adapt to different auxiliary samples, the model is discouraged from simply memorizing the limited training samples. (b) By randomly selecting auxiliary samples, we significantly increase the density and diversity of the training distribution. This directly addresses the problem of sparse training data in direct learning, making it more likely that the residual required for a given test sample (relative to its closest auxiliary sample) is within a space the model can generalize to.
>
> In summary, the increased difficulty is not beyond the capability of existing neural operators, but alleviating overfitting by discouraging memorization, and the implicit data augmentation creates a denser and more diverse training distribution that improves generalization. This is precisely why DeltaPhi delivers performance gains in data-limited settings, as the model learns a more robust and generalizable representation of the underlying system. The consistent improvements, even on the challenging Navier-Stokes problem, validate such benefits.
>
> > **[W4,Q2] Concerns on performance of DeltaPhi on chaotic systems.**
>
> We agree that extremely chaotic systems, where small input variations lead to huge output changes, present a difficult learning task. However, we would like to emphasize that this is a fundamental challenge for all data-driven neural operators, not a unique limitation introduced by DeltaPhi.
>
> Our core argument is that **within the general problems where direct neural operators are effective, DeltaPhi's residual learning framework, combined with its similarity-based retrieval mechanism, consistently improves generalization capability**. Our results empirically show that DeltaPhi achieves consistent gains on irregular/regular domain problems, various training numbers and cross-resolution scenarios over extensive base models.
>
> **Even for relatively complex systems like the Navier-Stokes equation, DeltaPhi remains its effectiveness.**
> (a) First, DeltaPhi has been equipped with explicit mechanism for eliminating the negative effects of chaotic properties in complex dynamics. By explicitly constraining the retrieval range $K$, the introduced similarity-based retrieval mechanism essentially limits the maximum difference between input pairs ($a_i,a_k$). This ensures the network learns from residuals ($u_i−u_k$) that are  manageable and less chaotic, making the residual learning feasible even when the system's stability is weaker (i.e. $C$ in Equation 2 is large).
> (b) In addition, our experiments show that even without problem-specific tuning of $K$ and using only a simple, single-step retrieval based on the input state, DeltaPhi still delivers performance gains across seven different backbone models (Table 2). This validates that learning residuals is a more effective strategy for generalization. Though beyond the core of current work, we believe performance could be further enhanced with more problem-specific designs, such as designing specific retrieval metric, re-retrieving auxiliary samples based on the most recent time step, etc.
>
> Based on your valuable suggestion, we include an additional comparison on the Navier-Stokes with relatively higher viscosity of 1e-3 and 1e-4. The output steps are 40 and 20 steps for viscosity 1e-3 and 1e-4, respectively.
>
> |Model|v=1e-4|v=1e-3|
> |---|---|---|
> |FNO|3.04e-1|1.21e-1|
> |FNO-DeltaPhi|2.43e-1|6.88e-2|
>
> The results validate the consistent gains on Navier-Stokes problem. We will include more base models in our final manuscript.
>
> > **[Q1] Performance without residual connection.**
>
> Thank you for the insightful question. While direct prediction seems intuitive, it has two main drawbacks compared to residual learning: (a) Failure to Utilize Auxiliary Input: The model is not effectively compelled to use the auxiliary information. Since the target output remains fixed for a given input, the network can simply learn to ignore the changing auxiliary sample, as it isn't necessary for producing the correct answer. (b) Lack of Label Diversity and Overfitting Risk: This approach fails to enhance training label diversity (the training labels are the same as in direct learning). It can lead the model to memorize the limited set of solutions, increasing the risk of overfitting. We provide the results in the table below:
>
> |Model|Relative Error|
> |---|---|
> |FNO|3.70e-2|
> |FNO-DeltaPhi (Residual Prediction)|3.31e-2|
> |FNO-DeltaPhi (Direct Solution Prediction)|4.38e-2|
>
> > **[Q3] Question on Table 4**
>
> We understand your insightful confusion regarding the trend of relative gains. However, the relative performance gain is influenced by several interacting factors, not solely the amount of training data. Key factors include the resolution of the training data, the specific generalization task, and the representational capacity of the base model.
>
> For instance, in the resolution generalization problem you referenced (Table 4), the model is trained on very low-resolution data but test on high-resolution data. In such a challenging scenario, the quality of the retrieved auxiliary sample becomes more critical for accurate residual prediction. Therefore, having a larger pool of data (900 samples vs. 100) significantly increases the likelihood of finding a suitable auxiliary sample, which in turn allows the residual learning framework to achieve a greater performance gain.
>
> In summary, the trend in relative gains can indeed vary depending on the specific experimental configuration. This does not alter our core conclusion: residual learning consistently alleviates the performance decline of neural operators in data-limited scenarios.
>
>
> > **[Q4] Experiment setting in Table 2**
>
> Yes, the training and testing resolution for Darcy flow in Table 2 are all $85 \times 85$.
>
> > **[Q5] Question on applicability of DeltaPhi.**
>
> Yes, DeltaPhi is a general framework that can be applied to all these neural operators. The reason we only reported the results of DeltaPhi with NORM in Table 1 is that the other baseline results were directly cited from the NORM [1]. Unfortunately, the official code for these baselines has not been open-sourced, which prevents us from conducting a fair and direct comparison by applying DeltaPhi to them.
>
> To demonstrate its broader compatibility, we further test DeltaPhi on additional models for the irregular darcy problem, including Transolver (ICML 2024) [2] and HPM (ICML 2025) [3]. The results are presented below. We plan to include more baseline comparisons on irregular problems in the final version of our paper.
>
> |Model|Relative Error|
> |---|---|
> |Transolver|8.54e-3|
> |Transolver-DeltaPhi|8.18e-3|
> |HPM|7.39e-3|
> |HPM-DeltaPhi|6.67e-3|
>
> **Reference**
> - [1] Learning neural operators on riemannian manifolds
> - [2] Transolver: A fast transformer solver for pdes on general geometries
> - [3] Holistic Physics Solver: Learning PDEs in a Unified Spectral-Physical Space
>
> > **[Minor Issue] Definition of neural operator.**
>
> Thank you for this valuable suggestion. We will revise Equation (1) in the final version to provide a more general formulation that is applicable to a broader class of neural operators like DeepONet series.

---

> > ### Comment · Reviewer_cdGa · 2025-08-03
> >
> > Thank you to the authors for the detailed response.
> >
> > Overall, my concerns have been addressed. After reading the other reviewers' comments and giving the paper careful consideration, I have decided to keep my score as is, primarily for the following two reasons:
> >
> > 1. The use of cosine similarity is limiting (I read the responses; however, I am not fully convinced), and there is no principled way to identify suitable similarity measures.
> >
> > 2. While I agree with the authors that chaotic systems are challenging for all neural operators, they may pose even greater difficulty for the proposed method. Regarding the results provided: 1e-3 is not chaotic at all. It represents an overly simple and smooth dataset, which favors models like FNO and its variants that are designed to operate on low-frequency signals. Also from your results, we can already see that the performance gain from less smooth dataset (1e-4) is more marginal compared to that from smooth dataset (1e-3).
> >
> > Overall, I believe this paper presents an interesting method for operator learning and offers valuable insights to the community. **I support the acceptance of this work.**

---

> > > ### Author Response · Authors · 2025-08-05
> > >
> > > We sincerely appreciate your insightful response and recognition of this work. Your suggestions have been instrumental in significantly improving this project, and we appreciate the time and effort you dedicated to these helpful comments.
> > >
> > > While not altering the core contributions of this study, we agree that your points regarding cosine similarity and highly chaotic systems are important considerations. Accordingly, we will more comprehensively discuss the potential constraints of cosine similarity in the limitations section. We will also explicitly address the complexities associated with highly chaotic systems, noting this as an important problem warranting further investigation.
> > >
> > > All your other suggestions will be incorporated into the revised manuscript to further strengthen the work.
> > >
> > > If you have other questions or concerns, please feel free to raise them. We will be more than willing to respond to them.

---

### Official Review · Reviewer_qC6c · 2025-06-20

**Clarity:** 3
**Significance:** 3
**Originality:** 3
**Rating:** 4
**Confidence:** 2

**Summary:**

This paper presents DeltaPhi, which is a novel residual learning framework for neural operator-based PDE solvers, and targets at scenarios where training data is scarce. Instead of learning direct input-output mappings, DeltaPhi learns residual mappings between pairs of similar physical states, leveraging the stability of PDE systems, which effectively augments the training distribution without additional data by generating multiple residual pairs. DeltaPhi is agnostic and compatible with various neural operators, and evaluated across regular and irregular domains, cross-resolution generalization, and varying data sizes.

**Questions:**

- For temporal PDEs (e.g., Navier-Stokes), the residuals are learned against earlier frames. In this case how does the error propagate over multi-step predictions?
- Can DeltaPhi scale to very large-scale systems where each sample is massive?

**Ethical Concerns:**

["NO or VERY MINOR ethics concerns only"]

**Limitations:**

Yes.

**Quality:**

3

**Strengths And Weaknesses:**

- Strengths:
  - Theoretical innovations: This paper reformulates PDE solving as residual learning between physical states. This formulation is well-grounded in physical principles (stability of PDEs).
  - Implicit data augmentation: The method increases the diversity of training targets without requiring new data, directly addressing data scarcity.
- Weaknesses:
  - Additional overhead from the retrieval process: While efficient, it still introduces extra complexity that may hinder deployment in some real-time applications.
  - Similarity metric: The Cosine similarity metric may not always align with physical relevance, especially for chaotic or nonlinear dynamics.
  - No Discussion on boundary conditions: DeltaPhi may behave differently for out-of-distribution samples, such as in systems with sharp boundary effects or discontinuities.

---

> ### Author Rebuttal · Authors · 2025-07-31
>
> Thanks for your thoughtful reviews and insightful questions. They have significantly improved this work. We respond point-by-point below.
>
> > **[W1] Concerns on efficiency of retrieval.**
>
> We would like to clarify that **the retrieval overhead from DeltaPhi is minimal and could be further enhanced for real-time applications**.
>
> First, our evaluation in Appendix C.2 has shown that the current implementation is highly efficient. The retrieval process introduces a negligible overhead of only 0.2–0.3 ms on a standard GPU, which is well within the acceptable limits for many real-time systems.
>
> Furthermore, for applications with even stricter latency requirements or larger datasets, our framework's scalability can be easily enhanced. Thanks to the robust research and development efforts [1,2], advanced retrieval tools can navigate vast amounts of data without a linear increase in retrieval time. For example, our framework can seamlessly incorporate well-supported, GPU-accelerated libraries like Faiss [3]. Faiss is an open-source library proven to retrieve vectors from billion-scale datasets in just a few milliseconds. By leveraging such tools, we can ensure DeltaPhi remains highly scalable and suitable for demanding, large-scale deployments.
>
> **Reference**
> - [1] Billion-scale similarity search with GPUsc
> - [2] Fast search in hamming space with multi-index hashing
> - [3] The faiss library
>
> > **[W2] Concerns on cosine similarity.**
>
> We'd like to clarify that **cosine similarity is effective in the systems we studied, including those with complex dynamics**. While the dynamics of such systems are complex over long durations, they exhibit sufficient stability over the shorter time intervals typically used in neural operator-based PDE solving. This short-term stability ensures that physically similar initial states lead to similar subsequent trajectories, which is the foundation for cosine similarity to be a meaningful measure. Our strong results on the Navier-Stokes equation empirically validate this.
>
> In addition, we agree that there is significant room for future research into optimal similarity metrics, especially for more complex physical systems or different problem settings. For instance, it is significant to explore performing retrieval in a latent space to capture more complex physical relationships. However, for the scope of this work, our results conclusively show that a simple and efficient metric like cosine similarity is sufficient to unlock the substantial benefits of residual learning.
>
> > **[W3] Discussion on boundary conditions.**
>
> Thank you for this valuable comment. The behavior of neural operators under varying boundary conditions, especially for out-of-distribution samples with sharp effects, is a critical aspect of PDE solving. We provide a discussion below and will include it in the final manuscript.
>
> While DeltaPhi does not explicitly process boundary conditions, its core mechanisms and our results demonstrate its effectiveness in handling systems with sharp boundary effects or discontinuities. (a) First, DeltaPhi explicitly leverages the full auxiliary solution as input, which provides a strong physical prior on the boundary region of complex systems. (b) In addition, our results on irregular domains provide strong evidence of DeltaPhi's robustness in handling complex geometries and their complex boundary effects. In problems like Pipe Turbulence and Blood Flow, the domains are irregular with complex inlet/outlet boundary conditions. As shown in Table 1, DeltaPhi achieves very significant relative gains of 40.99% and 11.00%, respectively. This demonstrates that the residual learning mechanism is effective even when faced with the complex boundary effects.
>
> We agree that extreme scenarios, such as generalizing to problems with extreme effects of boundary conditions, represent a challenging frontier for all neural operators. DeltaPhi could be potentially enhanced to better address this in the future. For example, the retrieval metric could be enhanced by designing a boundary-aware similarity metric that gives higher weight to boundary regions, ensuring a more proper auxiliary sample is chosen.
>
> > **[Q1] Questions about temporal pdes.**
>
> First, we wish to clarify that our framework does not learn the residual against previous frames of the same prediction sequence. Instead, it learns the residual between the target solution ($u_ i$) and a complete auxiliary trajectory ($u_{k_ i}$) retrieved from the training set.
>
> Regarding how error propagates in multi-step predictions, our method, like direct neural operators [1,2], is subject to error accumulation in auto-regressive rollouts. However, DeltaPhi provides a specific advantage in mitigating this effect. Specifically, residual learning approach helps stabilize the prediction by anchoring it to a known, physically valid trajectory ($u_{k_ i}$, which could be updated based on the status of last few time steps). Instead of predicting the full state from scratch at each step, the model only needs to learn the deviation from a plausible baseline. This is potentially beneficial for long-term predictions where direct methods can become unstable.
>
> **Reference**
> - [1] Fourier neural operator for parametric partial differential equations
> - [2] Transolver: A fast transformer solver for pdes on general geometries
>
> > **[Q2] Questions about large-scale systems.**
>
> While scaling to massive systems is challenging for general neural operators and beyond current work's core focus, DeltaPhi is potentially effective for such problems. (a) First, for systems with extremely high-resolution samples, similarity can be efficiently computed on downsampled data or important regions (such as boundaries), as core physical characteristics are preserved in lower-frequency components, bypassing the need to process the entire sample for retrieval. (b) In addition, for scenarios with a massive number of retrieval samples, the retrieval overhead is minimal, and the framework can integrate GPU-accelerated libraries like Faiss [1] to ensure millisecond-latency retrieval from billion-scale datasets. (c) The memory requirement for storing a large dataset is a general constraint in large-scale scientific computing and not a bottleneck unique to DeltaPhi. We will explore such large-scale problems in the future.
>
> **Reference**
> - [1] The faiss library

---

> > ### Comment · Reviewer_qC6c · 2025-08-05
> > **Response**
> >
> > Thank you for your rebuttal. My concerns are properly resolved. I'm keeping my original rating (weak accept).

---

> > > ### Author Response · Authors · 2025-08-06
> > >
> > > We sincerely thank you for your valuable feedback and thoughtful acknowledgment of our work. Your suggestions have been instrumental in helping us enhance this project significantly. We deeply appreciate the time you've taken to provide these helpful comments. If you have other questions or concerns, please feel free to raise them. We will be more than willing to respond them.

---

### Official Review · Reviewer_7n6g · 2025-06-24

**Clarity:** 2
**Significance:** 2
**Originality:** 2
**Rating:** 4
**Confidence:** 4

**Summary:**

This paper proposes DeltaPhi, a residual learning framework for neural operators that attempts to transform the PDE-solving task from directly predicting solutions to predicting residuals between similar physical states. The intuition behind this approach is interesting, and the authors substantiate their method with a variety of empirical studies across multiple PDE scenarios and data conditions. However, I do have several critical concerns arise regarding the mathematical rigor, foundational assumptions, and practical applicability of the proposed method.

**Questions:**

I have detailed my concerns in the weakness part. Below are some suggestions based on my concerns:
- Provide rigorous derivations or explicit constraints ensuring the Lipschitz continuity condition within their implementation.
- I suggest explicit analyses, empirical or theoretical, of the stability constant (C) to demonstrate its impact on the practical applicability of residual learning.
- Please establish clear mathematical conditions and proofs regarding the well-posedness and structural properties of the residual operator's codomain
- I suggest the authors to offer a detailed error analysis or theoretical justification for the use of interpolation methods in handling irregular or mixed-resolution data.

**Ethical Concerns:**

["NO or VERY MINOR ethics concerns only"]

**Final Justification:**

I’ve reviewed the responses and am satisfied with most of the explanations. I am good to increase my score from 3 to 4.

**Limitations:**

Yes

**Quality:**

2

**Strengths And Weaknesses:**

Strengths: The core concept of leveraging residual learning by exploiting the assumed stability of PDE solution operators is interesting. The extensive empirical experiments and benchmarks presented by the authors illustrate potential advantages of the proposed approach.

Weaknesses:
- The paper's main hypothesis relies heavily on the Lipschitz continuity of PDE solution operators. While this assumption holds for elliptic and parabolic PDEs with regular boundary conditions, its validity for hyperbolic or chaotic systems is questionable. For chaotic PDEs, initially similar states typically diverge rapidly due to exponential sensitivity. Also, the paper lacks rigorous mathematical derivations or the enforcement of hard constraints that would guarantee or at least ensure adherence to Lipschitz continuity within the model implementation.
- I do not see any explicit investigation or even mention the magnitude or potential variability of the stability constant (C). Given that a large or varying Lipschitz constant could dramatically amplify error propagation, undermining the very motivation of residual learning, it is crucial to address this aspect.
- The residual operator defined by the authors, $G^\Delta : A^2 \rightarrow \Delta U$, maps pairs of inputs to differences in solutions. However, the codomain $\Delta U = \{u_i - u_j \mid u_i, u_j \in U\}$ is not guaranteed to be well-defined or structured as a proper Banach space unless $ U $ itself is explicitly shown to be a linear space closed under subtraction. The authors implicitly assume $ \Delta U \subseteq U $ without proving or even explicitly stating the necessary conditions for this assumption to hold true.
- The subtraction $ u_i - u_j $ presupposes that both solutions $u_i $ and $ u_j $ are defined on identical grids or domains. In practice, this assumption is nontrivial, particularly for problems involving irregular geometries or mixed-resolution training sets. The authors address this via Fourier-based interpolation methods. Yet, interpolation inherently introduces approximation errors, particularly in sparse or irregularly sampled domains, and the paper provides no rigorous error analysis or guarantees regarding the accuracy or reliability of these interpolations.
- The proposed mthod relies on cosine similarity to select "similar" auxiliary inputs. However, cosine similarity is not invariant under general isometries and is only naturally meaningful within Hilbert spaces. Applying cosine similarity as a measure of proximity in more general function spaces is mathematically unjustified and potentially misleading.
- The authors assert that implicitly augmenting data through residual learning enhances training diversity. While intuitively plausible, this "simplification hypothesis" is presented without rigorous theoretical or mathematical support. There is no guarantee that learning residuals inherently simplifies the approximation task; indeed, it may inadvertently introduce complexity or even meaningless noise, making learning more challenging rather than simpler.

---

> ### Author Rebuttal · Authors · 2025-07-31
>
> Thanks for your thoughtful reviews and insightful questions. They have significantly improved this work. We respond point-by-point below.
>
> > **[W1,W2,Q1,Q2] Questions on Lipschitz continuity.**
>
> We deeply appreciate your insightful questions about the Lipschitz continuity, chaotic systems and stability constant $C$.
> In the following, we provide a systematic response to you in three parts (I, II and III).
> We will carefully enhance this in the final version to mitigate confusions.
>
> **Part I: Clarification: Lipschitz Continuity is a Foundation for General Nueral Operators, Not an Added Hypothesis for DeltaPhi.**
>
> Regarding your primary concern "DeltaPhi relies heavily on the Lipschitz continuity of PDE solution operators.", we wish to clarify a crucial point: **DeltaPhi does not introduce Lipschitz continuity as an additional or stronger assumption compared to standard direct operator learning.**
>
> In fact, the existence of a continuous mapping from input functions to solution functions (i.e., the stability of the PDE) is a fundamental prerequisite for *any* data-driven operator learning method to succeed [1], including the baselines we compare against. If small changes in the input could lead to arbitrarily large, discontinuous changes in the output, learning a generalizable mapping would be extremely challenging for neural networks [2].
>
> Our core contribution is not to enforce this property, but to propose a novel *training framework* that **more effectively leverages this inherent, pre-existing property** for improved data efficiency. While direct methods learn the mapping $a_i \rightarrow u_i$, we reformulate the task to learn the mapping $(a_i, a_j) \rightarrow (u_i - u_j)$. The feasibility of learning this residual mapping is founded on the same stability property that makes the original problem learnable. Our work demonstrates that this reformulation helps mitigate overfitting, especially in data-limited settings.
>
> **Reference**
> - [1] Universal approximation to nonlinear operators by neural networks with arbitrary activation functions and its application to dynamical systems
> - [2] Some fundamental aspects about lipschitz continuity of neural networks
>
> **Part II: Effectiveness of DeltaPhi on Complex Chaotic Problems.**
>
> You are correct that for chaotic systems, "initially similar states typically diverge rapidly," which implies that the Lipschitz constant $C$ can be large or grow over time. This is a well-known challenge for all long-term prediction models in such systems.
> However, this challenge affects **both direct operator learning and our residual learning framework equally.** It makes the underlying mapping harder to learn in general, but it does not invalidate our approach. The key question is whether DeltaPhi still provides a relative advantage over the baseline in these difficult scenarios.
>
> Our experiments on the **Navier-Stokes equation at a low viscosity of 1e-5 (a challenging regime)** provide a direct answer. Despite this challenge, **Table 2 shows that DeltaPhi consistently improves performance across all seven base models**. For FNO, we see a 4.86% gain, and for FFNO, an 8.54% gain. This demonstrates the robustness of DeltaPhi. Even when the stability property is weaker (i.e., $C$ is larger), learning residuals between similar states is still more effective than learning the direct mapping from scratch. Our framework's ability to provide gains even in this difficult setting strengthens our claim of general applicability.
>
> In summary, we do not claim that DeltaPhi solves the inherent difficulty of such complex systems, but rather that it is a superior training strategy that enhances the performance of existing operators even in these challenging cases.
>
> **Part III. Analysis of the Stability Constant's ($C$) Impact on Residual Learning**
>
> We agree that a large or varying stability constant $C$ makes the learning task more challenging. This is because it can amplify the magnitude and complexity of the output residual $u_i - u_j$ even for moderately different inputs $a_i$ and $a_j$.
>
> While $C$ is an intrinsic PDE property and often analytically intractable, our framework involves a practical tool to balance its effect: the retrieval range $K$. There is an inverse relationship between the difficulty imposed by $C$ and the optimal size of $K$. For a system with a larger $C$ (i.e., a more chaotic or less stable system), a smaller $K$ would be necessary to restrict sampling to only the most similar pairs, thereby keeping the learning task tractable. Conversely, for a more stable system with a smaller $C$, one can afford a larger $K$ to safely learn from more diverse samples and improve generalization.
>
> Our experiments empirically support this. The poor performance of random sampling (equivalent to an unconstrained $K$) in **Appendix A.1** shows that failing to manage the impact of $C$ harms performance. The same table shows that proper values of $K$ exist, balancing diversity and learning difficulty.
>
> Based on your valuable suggestion, we will add a detailed analysis of the stability constant $C$ and its relationship with the retrieval range $K$ to the final manuscript. For this work, we used $K=20$ for experimental consistency. We have provided extensive discussion on the influence of $K$, its impact on different test distributions, and alternative sampling strategies in Appendices A.1, A.2, C.1, and A.6.
>
>
> > **[W3,Q3] Concerns on definition of codomain.**
>
> We highlight that **the codomain for residual mapping is the same well-defined banach space used in direct operator learning**.
> This property follows directly from the foundational axioms of function spaces in PDE theory. Neural operator learning is premised on learning a mapping between Banach spaces, $\mathcal{G}: \mathcal{A} \to \mathcal{U}$. A key property of a Banach space $\mathcal{U}$ is that it is a complete, normed vector space. By the closure axiom of vector spaces, if any two functions $u_i$ and $u_j$ are elements of $\mathcal{U}$, their difference $u_i - u_j$ is also guaranteed to be an element of $\mathcal{U}$. Consequently, the set of all possible outputs of our residual operator is a subset of $\mathcal{U}$. Therefore, our residual learning task is well-posed, as it operates within the same well-structured Banach space as conventional neural operators, rather than mapping to an ill-defined space.
>
> We will include this in the final manuscript based on your helpful suggestion.
>
> > **[W4,Q4] Concerns on the error of interpolation methods.**
>
> We acknowledge that the interpolation operation inevitably introduces approximation errors. However, **the introduced error of Fourier interpolation does not hurt the core capability of neural operators in the resolution generalization problem**. For cross-resolution generalization, the base model, FNO [1] relies on preserving low-frequency information for prediction. Our interpolation method also precisely preserves these low-frequency components, while confining the approximation error to the high-frequency regions as the base model. While high-frequency error is inevitably introduced for both FNO and FNO-DeltaPhi, this specific error does not seriously hurt the model's stability on unseen resolutions. As the table below shows, our experimental results verify the importance of preserving low-frequency components, with Fourier interpolation outperforming other methods.
>
> |Interpolation Methods|Relative Error|
> |---|---|
> |Fourier Interpolation|6.91e-2|
> |Nearest Interpolation|7.39e-2|
> |Bilinear Interpolation|7.36e-2|
>
> In summary, the introduced error of Fourier interpolation is effectively constrained in the high-frequency component, thus not affecting the stable prediction in the resolution generalization problem.
>
> For your mentioned problem where "samples lie on non-identical geometric domains", DeltaPhi can be employed based on Geo-FNO [2]. It maps functions from various physical domains onto a single, unified grid. The residual connection would be directly performed in the unified grid, without the need for interpolation operation.
>
> **Reference**
> - [1] Fourier neural operator for parametric partial differential equations
> - [2] Fourier neural operator with learned deformations for pdes on general geometries
>
> > **[W5] Concerns on cosine similarity.**
>
> We agree that cosine similarity has limitations, especially in the scenario with all-zero vectors (described by Reviewer cdGa) where it is undefined.
>
> However, we wish to clarify that **cosine similarity is a simple but effective choice for problems studied in current experiments**. (a) The function spaces for the PDEs (e.g., Darcy Flow, Navier-Stokes, Heat Transfer) are typically Sobolev spaces, which are indeed Hilbert spaces, **making cosine similarity a mathematically sound choice, not unjustified and misleading** (concerned by Reviewer 7n6g). (b) More importantly, our empirical results confirm its practical effectiveness. The metric effectively identifies input functions that lead to similar solutions as validated in Figure 4. And the consistent performance gains across all experiments also demonstrate its suitability.
>
> Furthermore, our framework is not restricted to a single similarity function. As we show in Appendix A.5, other metrics such as Euclidean (L2 mentioned by Reviewer cdGa) and Manhattan distance also yield strong performance, demonstrating the flexibility of the DeltaPhi framework. We present the results below for convenience:
>
> | Similarity Functions | Relative Error |
> |--- |---|
> | Cosine Similarity|3.31e-2|
> | Euclidean Distance|3.31e-2|
> | Manhattan Distance|3.28e-2|
>
> We believe that exploring more advanced, domain-specific, or even learned similarity metrics is a promising direction for future research that could unlock further performance gains of DeltaPhi.
>
> > **[W6] Questions on the learning difficulty of residual mapping.**
>
> Please see our response in `[W3]` to Reviewer `cdGa`.

---

> > ### Comment · Reviewer_7n6g · 2025-08-04
> >
> > I’ve reviewed the responses and am satisfied with most of the explanations. I am good to increase my score from 3 to 4. I’ve also reviewed the other concerns, such as the learning difficulty of residual mapping and the computational cost, etc. Both the authors and future readers will benefit from these insights, so I encourage you to revise the manuscript and incorporate these points in the future.

---

> > > ### Author Response · Authors · 2025-08-05
> > >
> > > We are deeply grateful for your insightful feedback and recognition of our efforts. Your suggestions have played a crucial role in substantially improving this project. We truly value the time you've taken to provide these helpful comments.
> > >
> > > Following your advice, we will make comprehensive revisions to the final manuscript based on the discussions with all reviewers during the rebuttal period to strengthen our work.
> > >
> > > If you have other questions or concerns, please feel free to raise them. We will be more than willing to respond to them.

---

### Official Review · Reviewer_eyAE · 2025-07-04

**Clarity:** 4
**Significance:** 3
**Originality:** 2
**Rating:** 4
**Confidence:** 3

**Summary:**

The paper introduces a training paradigm for neural operators, DeltaPhi, based on residual learning between similar PDE solutions. The method makes the hypothesis that auxiliary trajectories are available and take advantage of the information provided, by learning the residual of the solution instead of the direct solution. Finally, the method is evaluated on several datasets and studied in different settings.

**Questions:**

-	What if no similar solution are avaiblable at inference ? How are performance evolving wrt similarity between samples ? My question here is how does a « bad » input function would worsen the output prediction ?
-	How does the method behaves on chaotic PDEs ? In such cases, eq 2 can be very sensitive, ie 2 PDEs with similar dynamics could exhibits very different behaviors from slightly different ICs.
-	How long does the method takes for training ? I saw the inference time comparison, but does it effect NO training ?
-	In lines 142-151/190-196, Have you tried other interpolation methods ? Like linear interpolation ? Direct downsampling ? Doesn’t the 0-padding changes the Fourier deocmposition of the function ? Have you tried other padding methods ?
-	In tab1, what does ‘-‘ means ?
-	Is the proposed method applicable to other architectures ?
-	In fig3, could you detail the experimental setting ? is it evaluated on $421\times421$ grids ?
-	In fig5 : how does the training set can be expanded while the testing set in smaller ? Is there a distribution shift in the datasets ?

**Ethical Concerns:**

["NO or VERY MINOR ethics concerns only"]

**Final Justification:**

DeltaPhi presents a new training paradigm for neural operator. The method is empirically evaluated on several standard datasets.
I agree with other reviewers and I will keep my score to 4.

I think additional studies on more challenging settings would convince the reader (eg on the meaning of the similarity between 2 samples, how it can be used in practice..), since it is the core of the method.

**Limitations:**

Yes a limitation section is provided.

**Quality:**

3

**Strengths And Weaknesses:**

### Strenght :
-	The paper is well-written and easy to follow
-	I found the idea interesting and experiment seems to show that the method effectively helps in learning operators.
-	Several experiment illustrate the claims

### Weaknesses :
-	The starting hypothesis is strong, meaning that one has access to some PDE solution even at inference.
-	Experiment could focus on more challenging datasets to deeply study the impact of the method on several contexts (see questions).

---

> ### Author Rebuttal · Authors · 2025-07-31
>
> Thanks for your thoughtful reviews and insightful questions. They have significantly improved this work. We respond point-by-point below.
>
> > **[W1] Concerns on access to some PDE solution at inference.**
>
> We would like to clarify that in data-driven PDE solving, the assumption of having access to some PDE solutions during inference is a common and low-cost practice, rather than a strong hypothesis.
>
> (a) First, the training of any neural operator requires a dataset of PDE solutions. These existing training data can be readily stored and utilized during the inference stage. This approach, which uses training data as a non-parametric memory at test time, is a common practice in other fields [1,2]. Therefore, we believe this is a standard and reasonable assumption.
>
> (b) In addition, the costs associated with this approach are well within acceptable limits for scientific computing applications. The storage requirement for the training examples is typically a few gigabytes. The time required to retrieve examples from the training set is also negligible, usually taking only a few seconds (see Section C.2).
>
> **Reference**
> - [1] Training data is more valuable than you think: A simple and effective method by retrieving from training data
> - [2] ResMem: Learn what you can and memorize the rest
>
> > **[Q1] Performance evolving wrt similarity between samples.**
>
> Thanks for your insightful question, our analysis in Appendix A.2 has directly demonstrated that **DeltaPhi is robust with respect to the similarity of auxiliary samples and performs well even on a "bad" input function**.
>
> (a) To evaluate performance on dissimilar samples, we systematically divided the test set into 7 non-overlapping splits based on the similarity score between each test sample and its closest training sample, as shown in the table below. Splits with lower average scores (Split1, Split2 and Split3) represent more challenging, out-of-distribution test cases, directly simulating the effect of a "bad" input function you mentioned.
>
> |Split|Sample Number|Min Score|Max Score|Average Score|
> |---|---|---|---|---|
> |Split1 (Most Dissimilar)|23|0.85|0.86|0.86|
> |Split2|14|0.86|0.88|0.87|
> |Split3|30|0.88|0.90|0.89|
> |Split4|37|0.90|0.91|0.90|
> |Split5|33|0.91|0.93|0.92|
> |Split6|43|0.93|0.95|0.94|
> |Split7 (Most Similar)|16|0.95|0.96|0.95|
>
> (b) As presented in the following table, the results demonstrate that DeltaPhi consistently outperforms direct operator learning across all splits, even the most challenging ones that deviate significantly from the training set:
>
> |Method|Split1 (Most Dissimilar)|Split2|Split3|
> |---|---|---|---|
> |Direct Learning|5.08e-2|5.00e-2|4.71e-2|
> |DeltaPhi ($K$=20)|4.77e-2 (**6.06%** $\uparrow$)|4.68e-2 (**6.34%** $\uparrow$)|4.32e-2 (**8.28%** $\uparrow$)|
>
> More results of different splits are shown in Appendix A.2. These results empirically demonstrate that even when the retrieved auxiliary sample is not a close match, DeltaPhi effectively predicts the solution residual.
>
> > **[W2,Q2] Effectiveness on challenging datasets involving chaotic properties.**
>
> We understand your mentioned "more challenging datasets" in "Weakness 2" means the complex problems involving chaotic properties, which are typically challenging for general neural operators [1,2]. While DeltaPhi does not solve the inherent difficulty of direct neural operators on such complex systems, **it is an effective training strategy that enhances the performance of existing operators even in these challenging cases**.
>
> To substantiate this, we have included the challenging Navier-Stokes equation (with low viscosity 1e-5) in the submitted manuscript.
> (a) For handling the complex problem, we follow previous works [2] and learn the system's evolution over a limited, short-term time interval where the mapping from input to output remains well-defined and predictable (i.e. similar inputs leading to similar outputs).
> (b) Table 2 present the results. For FNO, we see a 4.86% gain, and for FFNO, an 8.54% gain. This demonstrates the robustness of DeltaPhi. Even when the stability property is weaker (i.e., $C$ in Equation 2 is larger), learning residuals between similar states is still more effective than learning the direct mapping from scratch. Our framework's ability to provide gains even in this difficult setting strengthens our claim of general applicability.
>
> Therefore, compared to classical direct operator learning, DeltaPhi does not introduce excessive challenges for modeling chaotic systems.
>
> **Reference**
>
> - [1] Deeponet: Learning nonlinear operators for identifying differential equations based on the universal approximation theorem of operators
> - [2] Fourier neural operator for parametric partial differential equations
>
>
> > **[Q3] Training time cost of residual learning.**
>
> Residual operator learning does not introduce a significant training overhead compared to a standard neural operator.
> (a) Identical Training Steps: The number of training epochs and steps are kept exactly the same as in the direct neural operator baseline. During each training step, an input function is paired with a single, randomly chosen auxiliary sample. This enhances training diversity without requiring additional optimization steps.
> (b) Minimal Per-Step Cost: The additional cost introduced in each training step—sampling the auxiliary function, concatenation, and the residual connection—are computationally inexpensive.
>
> To provide a quantitative comparison, we report the average training time per epoch for the Darcy Flow experiment on an NVIDIA RTX 4090 GPU. The results are as follows:
>
> |Model|Training Time / Epoch (s)|
> |---|---|
> |FNO|0.265 ± 0.014|
> |FNO-DeltaPhi|0.309 ± 0.019|
>
> As the table shows, the increase in training time is minimal (only 0.05 second per epoch), confirming that DeltaPhi is efficient and comparable to the standard approach.
>
> > **[Q4] Questions on interpolation methods.**
>
> The employed Fourier-based interpolation is plausible and advantageous for cross-resolution solution prediction. Zero-padding in the frequency domain is equivalent to ideal sinc interpolation in the spatial domain, which effectively upsamples the signal while **exactly preserving the original low-frequency components without distortion**. This aligns with the core principles of the FNO architecture for zero-shot resolution generalization, i.e. accurately capturing the low-frequency dynamics of the physical solution.
>
> We tested other methods including Bilinear Interpolation and Nearest Interpolation on Darcy problem (training resolution is $85 \times 85$ and testing resolution is $421 \times 421$), as presented in the following table.
>
> |Interpolation Methods|Relative Error|
> |---|---|
> |Fourier Interpolation|6.91e-2|
> |Nearest Interpolation|7.39e-2|
> |Bilinear Interpolation|7.36e-2|
>
> The results show that Fourier-based interpolation performs better for zero-shot resolution generalization. We will include the analysis in our final manuscript.
>
> > **[Q5] The meaning of '-' in Table 1.**
>
> In Table 1, the symbol ‘-’ indicates that the baseline method could not be directly applied to the problem for the following reasons: (a) For the Heat Transfer and Blood Flow problems: GraphSAGE and FNO are not applicable because the input and output functions are defined on different physical domains. (b) For the Composite problem: FNO is not applicable because the physical domain is irregular.
>
> > **[Q6] Other architectures.**
>
> The formulated residual mapping is a general operator learning task, fundamentally independent of the specific base architecture. Consequently, **any neural operator can be employed to learn this mapping**. In the following table, we have conducted new experiments integrating our method with an additional architectures Transolver (ICML24) [1] and HPM (ICML25) [2], on Irregular Darcy problem.
>
> |Model|Relative Error|
> |---|---|
> |Transolver (ICML24)|8.54e-3|
> |Transolver-DeltaPhi|8.18e-3|
> |HPM (ICML25)|7.39e-3|
> |HPM-DeltaPhi|6.67e-3|
>
> The results show that DeltaPhi consistently improves the base models. We will include more base models across more problems in our final manuscript.
>
> **Reference**
> - [1] Transolver: A fast transformer solver for pdes on general geometries
> - [2] Holistic Physics Solver: Learning PDEs in a Unified Spectral-Physical Space
>
> > **[Q7] Experimental setting in Figure 3.**
>
> Yes, the "High Resolution" curves are at resolution $421 \times 421$.
>
> > **[Q8] Questions on Distribution Augmentation.**
>
> The expanded training set distribution and more concentrated testing set distribution observed in Figure 5 are a result of **the asymmetric sampling mechanism in the residual learning framework**.
> (a) During training, we leverage the stability property to augment the diversity of training samples. This is achieved by pairing each input with auxiliary samples from a proper range of similarities. This process effectively expands the training distribution, exposing the model to a wider variety of physically valid residuals and enhancing its ability to generalize.
> (b) During testing, however, our goal is to achieve the most accurate prediction for a given input. To do this, we could select the single most similar auxiliary sample from the training set. This focused approach leads to a more concentrated distribution of residuals for the test set, as each prediction is based on the closest available reference.
>
> This is the key advantage of residual learning framework, as it systematically improves the model's generalization performance by diversifying training while ensuring precision at inference. We will refine the explanation about this in final version to avoid any confusion.

---

> ### Comment · Reviewer_eyAE · 2025-08-02
> **Answer to Authors' Rebuttal**
>
> I apologize for the late response and I thank the authors for their clarifications, their answers to my questions, and the additional experiments. After reading the rebuttal, I now have the following additional questions and remarks:
>
> -	(Q1) Thanks for the additional experiment. Do you have any insight into what a similarity score of 0.8 between two samples represents in practice? How different are the samples at that level of similarity? Additionally, do you have any intuition about how the underlying PDE parameters influence the similarity between two samples? I do not necessarily ask for additional experiments here, but rather some intuition or explanation.
>
> In conclusion, I think the method is interesting and helps improving existing methods, I am keeping my score to 4 for acceptance.

---

> > ### Author Response · Authors · 2025-08-09
> >
> > We are deeply grateful for your insightful feedback and continued engagement. Your suggestions have played a crucial role in substantially improving this project, and we truly value the time you've taken to provide these helpful comments.
> >
> > Here is our response for your additional questions:
> >
> > **(a) On the practical understanding of similarity scores:**
> > We understand your interest in an intuitive understanding of the similarity values. While this is hard to visualize precisely, we provide some analysis based on previous results. As shown in Table 7 (Appendix A.2), when the average similarity score between the test sample and its most similar training sample drops from `0.95` (in Split 7) to `0.86` (in Split 1), the baseline model's prediction error more than doubles (from `2.43e-2` to `5.08e-2`). **This demonstrates that for the Darcy Flow problem, a similarity score of 0.8 already represents a significant functional distance**, where the system's behavior is substantially different. It's worth noting that, even in such a challenging, out-of-distribution scenario, DeltaPhi still delivers valid performance gains, highlighting its effectiveness.
> >
> > **(b) On the influence of underlying PDE parameters:**
> > You are correct that the relationship between similarity scores and functional distance is influenced by the underlying PDE and its parameters. This can be understood through the lens of the system's stability, as defined by the Lipschitz condition in Equation 2: $\|\mathcal{G}(a_ 1) - \mathcal{G}(a_ 2)\|_ {\mathcal{U}} \leq C\|a_ 1 - a_ 2\|_ {\mathcal{A}}$. The constant `C` quantifies the system's sensitivity. For stable systems (e.g., Navier-Stokes with high viscosity parameters), `C` is small. A small difference in inputs `a(x)` leads to a proportionally small difference in solutions `u(x)`. For more sensitive or chaotic systems (e.g., Navier-Stokes with low viscosity parameters), `C` becomes large. **Therefore, for the same similarity score between input functions, a system with more challenging PDE parameters (like a lower viscosity) will exhibit a larger functional distance between output solutions**. This is why the retrieval mechanism in DeltaPhi is crucial. As noted in our response to `W2` of Reviewer 7n6g and `W4` of Reviewer cdGa, the bounded retrieval range `K` helps avoid pairing samples that are excessively dissimilar in challenging systems. This ensures the residual learning task is tractable and effective. Our results on Navier-Stokes (with low viscosity 1e-5) also show that DeltaPhi is effective even on challenging PDE parameters.
> >
> > We hope this analysis provides the clarity and intuition for your questions, and we thank you again for your valuable engagement. We would be more than happy to address any further questions you may have.

---

> > > ### Comment · Reviewer_eyAE · 2025-08-09
> > > **Answer to Authors' Rebuttal (2)**
> > >
> > > I thank the authors for these additional answers and for the intuitions about the similarity between samples.
> > > This will be take into consideration during the discussion phase with other reviewers and AC, if needed.

---

> > > > ### Author Response · Authors · 2025-08-09
> > > >
> > > > Thank you for your response. Your involvement has greatly helped us improve this work. We will refine the final version based on your helpful suggestions to avoid misunderstandings.

---

### Decision · Program_Chairs · 2025-09-17

**Decision:**

Accept (poster)

**Comment:**

The paper addresses an important challenge in data-driven PDE solving: neural operators need large, high-quality datasets, which are often expensive or unavailable. Direct input–output mapping in such regimes limits generalization. The authors build on the observation that many physical systems are stable, with nearby initial states leading to nearby solutions. They reformulate the task so that the model learns residuals between a target solution and an auxiliary one. This reframing reduces overfitting, increases training diversity without new data, and better exploits limited information.

The main contribution is the DeltaPhi framework, an architecture-agnostic residual learning strategy that can be combined with standard operators such as FNO, FFNO, or more recent solvers. Experiments show that DeltaPhi improves generalization in data-limited settings, across both regular and irregular domains, and in cross-resolution transfer, with only minimal computational overhead. Gains are consistent though not always large, including on Navier–Stokes. The contribution lies in providing a general training paradigm rather than a new operator design.

In conclusion, the paper has some weaknesses, such as the simple similarity metric and limited gains on chaotic systems, but the contribution is clear. The framework is original, addresses an important problem, and is supported by solid experiments and clarifications during rebuttal. Reviewers converged on the view that the strengths outweigh the weaknesses. I recommend acceptance.